# Kupffer cells determine intrahepatic traffic of PEGylated liposomal doxorubicin

Kuan Jiang ®[1,2] ✉, Kaisong Tian[2], Yifei Yu[2], Ercan Wu[2], Min Yang ®[2], Feng Pan[3], Jun Qian[3] & Changyou Zhan ®[2,3] ✉

Intrahepatic accumulation dominates organ distribution for most nanomedicines. However, obscure intrahepatic fate largely hampers regulation on their in vivo performance. Herein, PEGylated liposomal doxorubicin is exploited to clarify the intrahepatic fate of both liposomes and the payload in male mice. Kupffer cells initiate and dominate intrahepatic capture of liposomal doxorubicin, following to deliver released doxorubicin to hepatocytes with zonated distribution along the lobule porto-central axis. Increasing Kupffer cells capture promotes doxorubicin accumulation in hepatocytes, revealing the Kupffer cells capture-payload release-hepatocytes accumulation scheme. In contrast, free doxorubicin is overlooked by Kupffer cells, instead quickly distributing into hepatocytes by directly crossing fenestrated liver sinusoid endothelium. Compared to free doxorubicin, liposomal doxorubicin exhibits sustained metabolism/excretion due to the extra capture-release process. This work unveils the pivotal role of Kupffer cells in intrahepatic traffic of PEGylated liposomal therapeutics, and quantitively describes the intrahepatic transport/distribution/elimination process, providing crucial information for guiding further development of nanomedicines.

Lipid-based nanocarriers have been widely utilized for the delivery of both small molecules and biomacromolecules for several decades[1–3]. From Doxil® to COVID-19 mRNA vaccines, there are nearly thirty kinds of marketed lipid-based nanomedicines worldwide[4–6]. Although with a long history, lipid-based nanocarriers in clinics are still mainly restricted to the PEGylated liposomes (sLip), which should be traced back to the last century[7]. For innovative lipid-based nanocarriers, such as thermo-responsive liposomes aiming to improve drug release, or surface functionalized nanocarriers such as actively targeting liposomes or micelles, are still without successful clinical translation[8,9]. Difficulty in clinical translation of next-generation lipid-based nanocarriers should be mainly ascribed to the ambiguous in vivo fate, which largely hampers the rational design in vitro and efficient regulation in vivo.

Intravenous injection is the preferred administration route for nanomedicines both in clinic and laboratory research, considering the widespread property of nanoparticles from blood circulation. However, from systemic circulation to the lesion sites, those nanoparticles would suffer a lot, including plasma protein adsorption, blood vessel extravasation, mononuclear phagocyte system (MPS) endocytosis, penetration across different barriers and intracellular drug release, usually causing an unpredictable in vivo performance, especially compromised bioavailability and unexpected immune responses[10–13]. For thermo-responsive liposomal doxorubicin (ThermoDox®) designed for improved drug availability in tumor, there was little improvement in the initial Phase III clinical trial by combination with radiofrequency ablation (RFA) for the treatment of hepatocellular carcinoma in comparison to RFA alone[8]. Folic acid (FA) is a typical targeting ligand to folate receptor-α and has been modified on various nanocarriers for tumor-targeting drug delivery but still with unsatisfying efficiency in vivo. Recently, it was reported that FA-modified

[1]Eye Institute and Department of Ophthalmology, Eye & ENT Hospital, Fudan University, Shanghai 200030, P.R. China. [2]Department of Pharmacology, School of Basic Medical Sciences & Department of Pharmacy, Shanghai Pudong Hospital & State Key Laboratory of Molecular Engineering of Polymers, Fudan University, Shanghai 200032, P.R. China. [3]School of Pharmacy, Fudan University & Key Laboratory of Smart Drug Delivery (Fudan University), Ministry of Education, Shanghai 201203, P.R. China. ✉e-mail: jiangkuan@fudan.edu.cn; cyzhan@fudan.edu.cn

liposomes would induce heavy natural IgM adsorption on the surface, depriving receptor recognition ability and accelerating blood clearance in vivo, eventually an impaired distribution in tumors even less than unmodified liposomes[14]. In short, well-designed nanocarriers would interact with various biomolecules/cells in vivo and displayed an uncontrollable performance due to complicated regulatory effects, and as a result, an in-depth understanding about in vivo fate of nanocarriers is indispensable for the future development.

The importance of intrahepatic disposition for intravenously injected nanocarriers has been widely accepted, as 30% to 99% of injected nanocarriers would accumulate in the liver, which might be a determining factor for drug distribution in targeting sites, metabolic burden of liver cells, and even hepatotoxicity[15]. The liver is an important immune and metabolic organ, occupied by abundant cells belonging to MPS and responsible for eliminating foreign substances (for example, injected nanoparticles) from the bloodstream[16,17]. As for cells composition of the liver, more than 60% is hepatocytes (HC), followed by liver sinusoidal endothelial cells (LSEC, 10–20%), Kupffer cells (KC, about 5%), lymphocytes, etc., defined as nonparenchymal cells (NPC)[18]. Although with some progress in nondegradable inorganic

nanoparticles[19,20], the intrahepatic behavior of lipid-based nanocarriers is still indistinct, largely due to the intrahepatic degradability of both nanocarriers and encapsulated drug. In this study, intrahepatic traffic of PEGylated liposomal doxorubicin (sLip/Dox) was quantitatively studied to explore the intracellular transport, metabolism, and excretion in detail, especially to understand the influence of liposomal formulation on intrahepatic fate of the payload.

## Results

### Kupffer cells dominate intrahepatic capture of liposomes

Circulating nanoparticles tend to interact with various meeting cells, and the cellular uptake mainly depends on cell phenotypes and contact opportunities for the same nanoparticles. To rank the capacity of several major hepatic cells responsible for liposomes capture, 5-carboxyfluorescein (FAM) labeled PEGylated liposomes (FAM-sLip, in which FAM was chemically conjugated with liposomes, detailed information in Methods) were prepared to evaluate cellular uptake of intact liposomes in the liver (Fig.1a and Supplementary Table 1). As shown in Supplementary Fig. 1, plasma concentration of FAM-sLip continuously decreased from 1 to 48 h after administration, with a half-

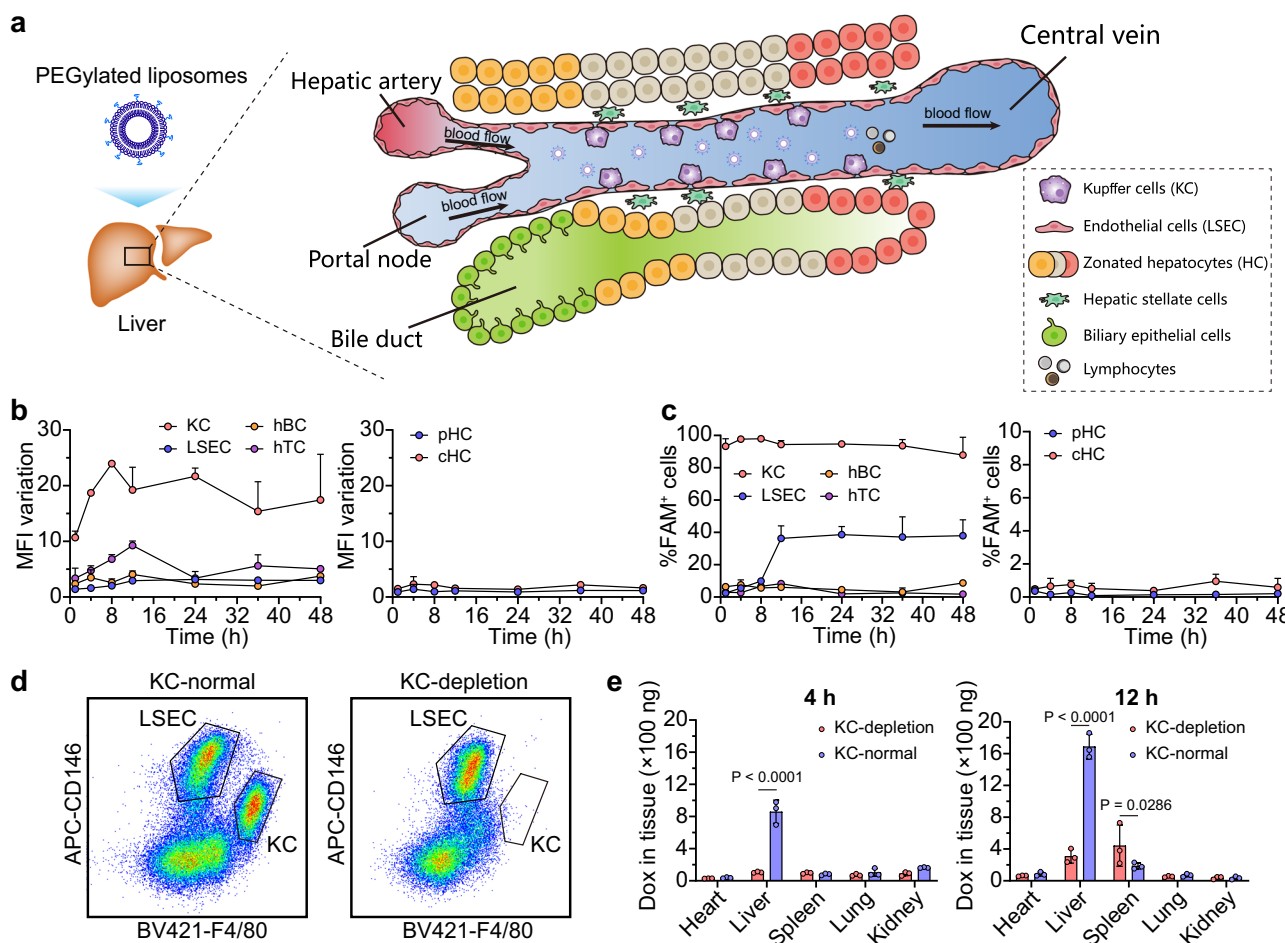

**Fig. 1 | Intrahepatic distribution of PEGylated liposomes. a** Intravenously injected PEGylated liposomes interact with various hepatic cells during flowing through the liver sinusoid. **b** Mean fluorescence intensity (MFI) variations and **c** the positive ratios of major hepatic cells interacting with 5-carboxyfluorescein (FAM) labeled PEGylated liposomes (FAM-sLip). The MFI variations were calculated as increasing folds compared to untreated cells. The positive ratios of untreated cells were gated as less than 0.1%. KC, LSEC, hBC, hTC, pHC, and cHC indicated Kupffer cells, liver sinusoidal endothelial cells, intrahepatic B lymphocytes and T lymphocytes, periportal hepatocytes, and pericentral hepatocytes, respectively. FAM-sLip was intravenously injected at a dose of 50 mg/kg HSPC. **d** Flow cytometry diagram

of nonparenchymal cells in mice liver. The KC was depleted by intravenously injecting clodronate liposomes at a volume of 200 µL and cellular composition was gated at 2 days after injection. The cell types were defined as KC (F4/80[+]) and LSEC (CD146[+]). **e** Tissues distribution of doxorubicin encapsulated in liposomes (90 nm) at 4 or 12 h post injection with KC depletion or not. The PEGylated liposomal doxorubicin (sLip/Dox) was intravenously injected at a dose of 5 mg/kg Dox. The statistical significance was analyzed by two-way ANOVA multiple comparisons corrected by Sidak's test. $P$ values are provided when there are statistical significances ($P < 0.05$). Data were means ± SDs for **b**, **c**, **e** ($n = 3$ mice). Source data are provided as a Source Data file.

life of 17.7 h, and area under the curve (AUC$_{1-48h, HSPC}$) about 7.00·mg·mL$^{-1}$ h. The major hepatic cells, including KC, LSEC, B lymphocytes (hBC) and T lymphocytes (hTC), and zonated HC along the lobule porto-central axis, were sorted by flow cytometry after labeling with corresponding biomarkers combined with density gradient centrifugation (Supplementary Figs. 2–4). As shown in Fig. 1b, c, KC exhibited quick and extensive endocytosis of FAM-sLip, with an increase of mean fluorescence intensity (MFI) about 10 to 25 folds, and more than 90% exhibited as FAM-positive from 1 to 48 h after injection compared to the untreated KC. The FAM-sLip uptake by LSEC was obviously different to KC although those two cells could directly interact with liposomes. Both the MFI and FAM-positive ratio of LSEC kept low within 8 h after administration. However, the MFI suddenly achieved a 3-fold increase in comparison to the untreated LSEC at 12 h after administration, and FAM-positive cells maintained at 40% until 48 h after administration, namely a lagging uptake behavior on FAM-sLip. For hBC and hTC, although with similar or even higher MFI increase compared to LSEC, the positive cell ratios kept lower than 10% from 1 to 48 h. In comparison to KC and LSEC, FAM-sLip distribution in HC was negligible, as MFI was comparable to untreated cells and the positive cells ratio kept less than 2%.

To further testify the importance of KC on liposome capture, sLip/Dox accumulated in different organs with KC depletion or not (Fig. 1d) was further evaluated. Compared to the normal condition (Fig. 1e), Dox accumulation in the liver was significantly reduced (about 1/8 at 4 h while 1/5 at 12 h of that in normal liver, $P < 0.0001$) when KC was depleted by intravenously injecting clodronate liposomes. But in other organs, Dox accumulation in normal or KC-depletion conditions was approximate, probably due to a much larger number of liver resident KC compared to other organs[17].

**Both liposome encapsulation and surface PEGylation dramatically accelerates the payload exposure to KC**

Intrahepatic cellular distribution of intravenously injected sLip/Dox, non-PEGylated liposomal Dox (Lip/Dox) and free Dox was compared to clarify carrier effects on intrahepatic fate of the payload. The sLip/Dox was reported with the high stability in blood circulation (more than 98% kept intact within 24 h after injection)[21], and prepared as shown in the Methods section and characterized as in Supplementary Table 1 and Supplementary Fig. 5. In blood circulation, as shown in Supplementary Fig. 6, sLip/Dox exhibited a highest AUC$_{1-24 h, Dox}$ compared to the other two preparations (1424.0 ± 116.0 for sLip/Dox, 1006.0 ± 28.9 for Lip/Dox, and 2.3 ± 1.2 µg·mL$^{-1}$·h for free Dox). Besides, as shown in Supplementary Fig. 7, blood cell counts were comparable between untreated and sLip/Dox treated mice, except for the lower number of white blood cells in treated mice, but still in the normal range (3.2–12.7 × 10$^3$/µL). To evaluate liver function after treatment by sLip/Dox, serum biochemical analysis was implemented and indicated no significant difference compared to that in untreated mice (Supplementary Fig. 8). From the hematoxylin-eosin staining sections of the liver in untreated or treated mice (Supplementary Fig. 9), there was also no obvious toxicity as both nucleus and cytoplasm were clearly stained. From the above results, single injection of sLip/Dox at a dosage of 5 mg/kg Dox would induce no toxicity for liver function or cell counts, supporting the following evaluation by this dosage.

As for intrahepatic cellular distribution, free Dox treatment induced the Dox concentration under the limit of detection (displayed as zero) from 1 to 24 h both in KC and LSEC after administration (Fig. 2a, b), indicating the small molecule was overlooked by those two cells. In contrast, two liposomal Dox resulted in significant Dox distribution in both two cells. For sLip/Dox, Dox concentration in KC was 62.5 ng/10$^6$ cells at 1 h, and maintained from 11.8 (4 h) to 17.1 ng/10$^6$ cells (24 h). Notably, FAM-sLip capture by KC resulted in constantly high fluorescence positive cell ratios from 1 to 48 h post injection, but Dox in liposomes exhibited a quickly decreasing distribution in

KC (Fig. 1c), suggesting that sLip/Dox capture by KC could be degraded to release Dox for further intrahepatic transport, meanwhile, the degraded liposomes could be delayed in cells. For non-PEGylated Lip/Dox, Dox distribution in KC was hysteretic to the PEGylated one, and displayed an increasing trend from 1 to 12 h, indicating differences in KC capture of two different liposomes, largely due to the existence of PEG molecules. Both liposomal Dox displayed a slow and moderate distribution behavior into LSEC. After 1 h, no detectable Dox was measured in LSEC, but with a continuous increase from 4 to 24 h, ranging from 0.1 to 2.9 ng/10$^6$ cells, which was less than that in KC but significantly higher than free Dox treated LSEC, especially after 12 h.

Although nearly no Dox entered KC and LSEC in free Dox-treated mice, there was quite a lot in HC at 1 h compared to liposomal Dox-treated mice. As shown in Fig. 2c, free Dox injection resulted in 3.06, 1.37, and 2.55 ng/10$^6$ cells of Dox in cHC, mHC, and pHC, respectively, which were over 8.5 times more than sLip/Dox treated mice, and 31.2 times more than Lip/Dox treated mice in zonated HC ($P < 0.001$), respectively. From 4 to 24 h (Fig. 2d–f), Dox distribution in HC from free Dox-treated mice suddenly decreased to be much lower than liposomal Dox-treated HC, indicating a quick infusion and fast excretion of free Dox in the liver, while a constant drug infusion and slower exit in the liver for liposomal Dox. Besides, in liposomal Dox-treated mice, Dox distribution in HC from different lobule zones also displayed a zonated distribution profile from cHC to pHC, as Dox concentration in pericentral HC was roughly much higher than that in HC from the other two regions. More importantly, as Dox distribution in liver cells ranking as KC » zonated HC for liposomal Dox treated group (Fig. 2g, h, but reversed in free Dox treated group as shown in Fig. 2i), indicating an intercellular transport from KC to HC for Dox encapsulated in liposomes. As sLip/Dox induced a more fast Dox distribution in KC than Lip/Dox, peaking at 1 or 12 h, respectively, and there was also slower Dox transport to HC in Lip/Dox treated group, also indicating a potential intercellular transport of liposomal Dox from KC to HC.

**Kupffer cells promote intercellular transport of sLip/Dox in the liver**

According to the microarchitecture of the liver (Fig. 3a), HC was spatially separated from circulating sLip/Dox by NPC (KC, LSEC, etc.) in the liver sinusoid. Considering liposomal Dox distribution in NPC and HC exhibited an intriguing concentration gradient, as in Fig. 2, the intercellular transport of sLip/Dox in the liver was further evaluated in this part. As for intracellular distribution, sLip/Dox with larger size accumulated in KC significantly more than the smaller one, as the larger sLip/Dox induced 1.5 to 3 folds higher Dox accumulation than the smaller counterpart (Fig. 3b). In LSEC, there was a sudden decrease for sLip/Dox accumulation compared to that in KC, ranging from 1 to 4 ng/10$^6$ cells in KC-normal state (Fig. 3c, d). Notably, for 90 nm sLip/Dox treated LSEC with KC depletion at 4 h, Dox concentration increased to 3.6 ng/10$^6$ cells, significantly more than in the normal condition, revealing a probable shielding effect of KC on LSEC for liposomes capture in short time. However, the situation reversed for 300 nm sLip/Dox, as Dox concentration in LSEC with KC depletion was 2.2 times lower than that in KC-normal state, probably due to LSEC uptake of nanoparticles with a cut-off size no more than 300 nm.

According to Fig. 3e and Supplementary Fig. 10a, Dox concentration in cHC and mHC of normal mice (without KC depletion) was higher than that in KC-depletion state at 4 and 12 h, especially in cHC, about 10.7 and 8.0 times more ($P < 0.01$), respectively. However, Dox concentration in pHC was close at 4 or 12 h under both conditions, indicating that KC mainly promoted Dox distribution in cHC but not in pHC to achieve zonated distribution. It was also exhibited that KC depletion induced comparable Dox accumulation in cHC, mHC, and pHC, namely a loss of zonated drug distribution in HC due to the absence of KC (Fig. 3f and Supplementary Fig. 10b). Similar to the results in KC, liposomal Dox with a larger particle size also

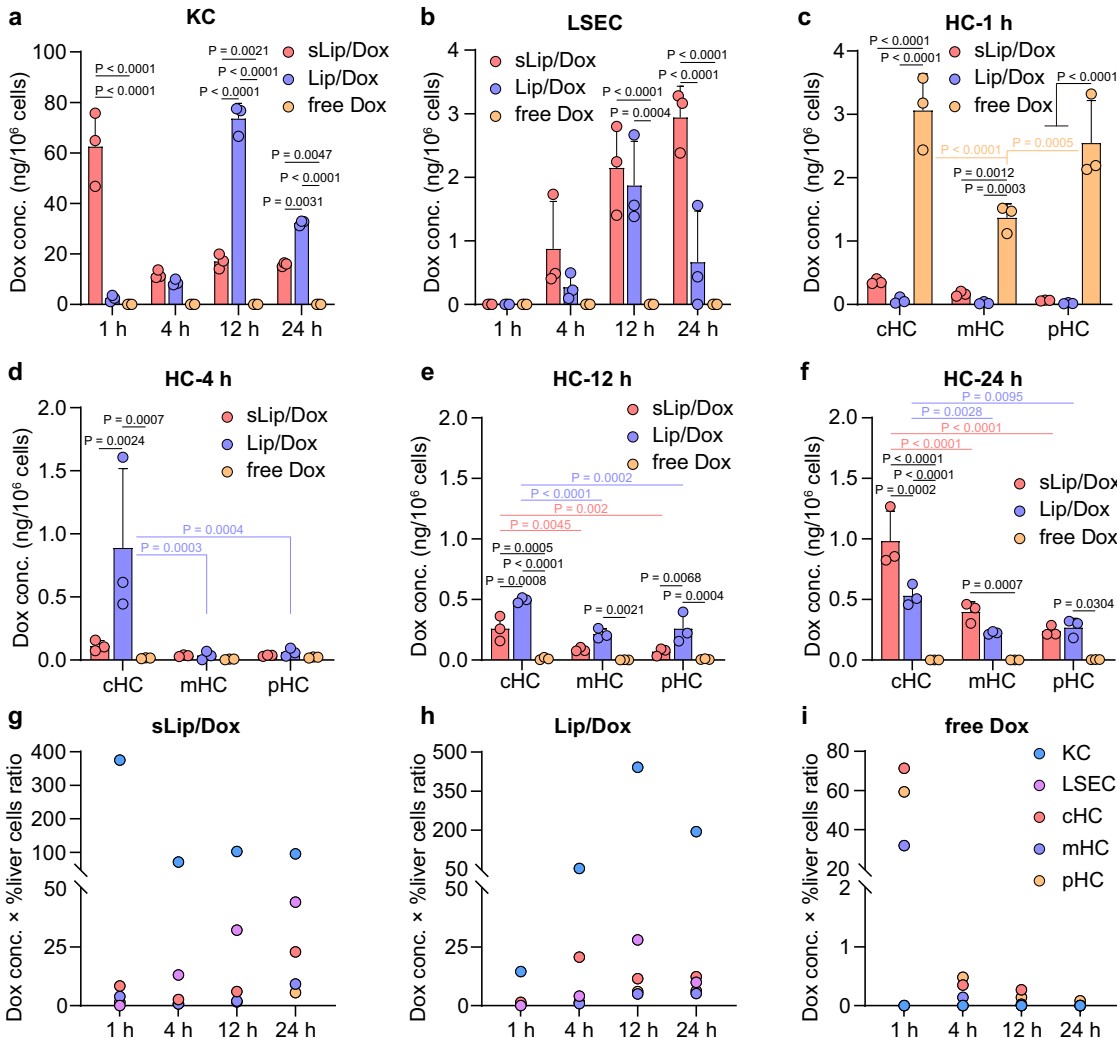

**Fig. 2 | Intrahepatic cellular transport of doxorubicin (Dox) in PEGylated liposomes (sLip/Dox), non-PEGylated liposomes (Lip/Dox), or in free form.** Dox concentration in Kupffer cells (KC, **a**), liver sinusoidal endothelial cells (LSEC, **b**), and zonated hepatocytes (HC, **c**–**f**) at 1, 4, 12, and 24 h post intravenous injection of liposomal Dox or free Dox at a dose of 5 mg/kg Dox. Rough calculation of total Dox distribution in major liver cells from mice treated by sLip/Dox (**g**), Lip/Dox (**h**), or free Dox (**i**). The Dox concentration was reused from data in **a**–**f**. The %liver cells ratio referred to ref. 18, as 6% for KC, 15% for LSEC, and 23.3% for cHC, mHC, and pHC, respectively, considering a total HC occupied about 70% of the total liver cells. cHC, pericentral hepatocytes, mHC, mid-lobule hepatocytes, and pHC, periportal hepatocytes. The statistical significance was analyzed by two-way ANOVA multiple comparisons corrected by Sidak's or Tukey's test. $P$ values are provided when there are statistical significances ($P < 0.05$). Data were means ± SDs for **a**–**f** ($n = 3$ mice). Source data are provided as a Source Data file.

accumulated more in HC compared to the smaller one (Fig. 3g and Supplementary Fig. 10c), especially in cHC, revealing a potential relationship of KC capture and Dox accumulation in HC. Differences in cellular accumulation were also reflected in plasma pharmacokinetic profiles of sLip/Dox (Supplementary Figs. 6, 11). Plasma concentration of Dox in KC-depletion mice was continuously higher than that in KC-normal mice after injection of liposomal Dox, indicating a constant contribution of KC for their blood clearance. Comparing different sizes of liposomal Dox total distribution in liver cells (Fig. 3h and Supplementary Fig. 12), Dox accumulation was ranked as KC > LSEC > cHC > mHC > pHC, and increased with enlarged particle sizes, further to testify KC initiated and promoted liposomal Dox zonated distribution in HC.

Accordingly, a possible intercellular transport pathway was depicted as in Fig. 3i. It was deduced that sLip/Dox in the liver sinusoid would be firstly captured by KC to initiate the following process. Considering sLip/Dox accumulated in HC was significantly decreased in the KC-depletion state, it would be difficult for intact sLip/Dox across liver sinusoid endothelium to reach HC. Consequently, sLip/Dox tended to be destroyed in KC to release free Dox, which would be overlooked by LSEC but quickly infused into the HC through fenestration on the liver sinusoid endothelium, indicating a probable intercellular diffusion as KC capture → intracellular sLip/Dox degradation and Dox release → released Dox overlooked by LSEC/infusion into HC through fenestration → zonated accumulation of Dox in HC along the lobule porto-central axis. Indeed, intrahepatic accumulation of lipid-based nanomedicines such as sLip/Dox could be regulated by two strategies (Fig. 3j). Firstly, physicochemical properties modification, such as increasing particle sizes or modification of KC targeting ligands (such as RGD peptide)[22] would promote KC capture and upregulate intrahepatic accumulation. Secondly, adjustment on KC function to avoid KC capture would impede intrahepatic payload accumulation.

## Liposomes encapsulation slows doxorubicin metabolism/excretion

In blood circulation, sLip/Dox was quite stable as lower than 2% Dox was released from the liposomes[21]. In other words, degradation and metabolism of liposomal Dox mainly happened in the liver, but the metabolites could also return to peripheral blood (Supplementary

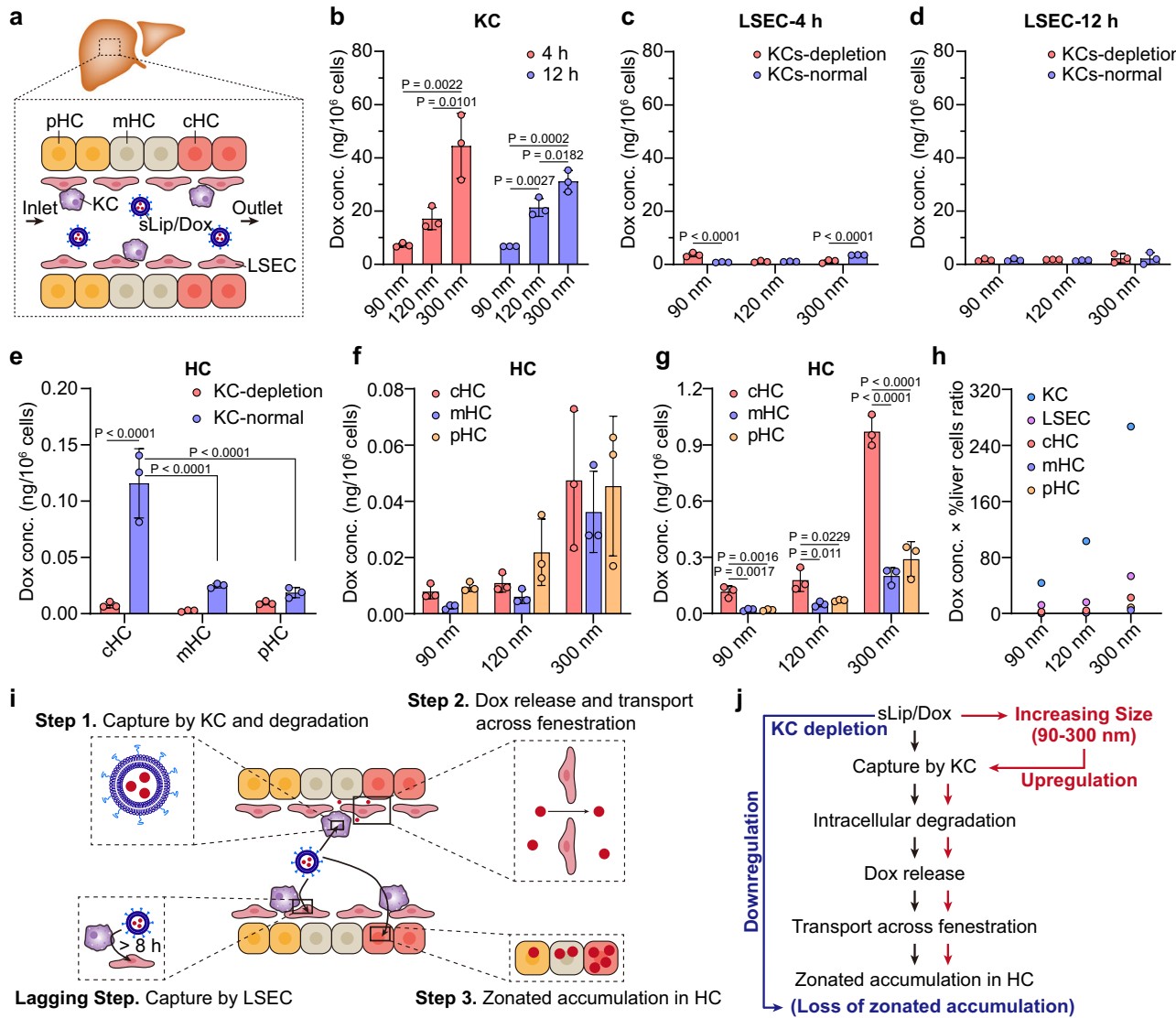

**Fig. 3 | Mechanism exploration for intercellular transport of sLip/Dox in liver.**
**a** Diagram of sLip/Dox interaction with various liver cells. The sLip with different particle sizes (90, 120, or 300 nm) encapsuled Dox distribution in KC (**b**) and LSEC (**c**, **d**) at 4 or 12 h post injection with KC depletion or not. **e** Distribution of sLip/Dox (90 nm) in zonated HC at 4 h post injection with KC depletion or not. **f** Distribution of sLip/Dox (90, 120, or 300 nm) in zonated HC at 4 h post injection with KC depletion. **g** Distribution of sLip/Dox (90, 120, or 300 nm) in zonated HC at 4 h post injection in KC-normal condition. The sLip/Dox was intravenously injected at a dose of 5 mg/kg doxorubicin. The statistical significance was analyzed by one-way ANOVA multiple comparisons corrected by Tukey's test for (**b**, **f**, **g**), and by two-way ANOVA multiple comparisons corrected by Sidak's test for (**c**–**e**). *P* values are provided when there are statistical significances (*P* < 0.05). Data were means ± SDs

for (**b**–**g**) (*n* = 3 mice). **h** Rough calculation of total Dox distribution in major liver cells at 4 h after injection. The %liver cells ratio referred to ref. 18, as 6% for KC, 15% for LSEC, and 23.3% for cHC, mHC, pHC, respectively, considering a total HC occupied about 70% of the total liver cells. **i** The schematic of intercellular transport pathway for sLip/Dox in liver. For intravenously injected sLip/Dox, firstly KC captured the liposomes and resulted in intracellular degradation, then Dox released extracellularly and transported across liver sinusoid endothelial fenestration into HC, and zonally accumulated in HC along the lobule porto-central axis. LSEC capture for sLip/Dox was less than that in KC and probably shielded by KC. **j** Regulatory effects of particle sizes variation or KC function on HC accumulation of Dox encapsuled in PEGylated liposomes. Source data are provided as a Source Data file.

Fig. 13). Considering no metabolite was detected in KC and LSEC, although there was more Dox accumulation in these two populations of cells, HC might be the main site for Dox metabolism in liver as various drug metabolic enzymes distributed in HC[23]. To explore the contribution of zonated HC to Dox metabolism, the distribution profile of doxorubicinol (Doxol, the main metabolite of Dox with pharmacological activity) was evaluated as in Fig. 4a. Within 8 h after administration, Doxol concentration in HC kept under the limit of detection. However, Doxol concentration in cHC was higher than that in mHC and pHC from 12 to 48 h, revealing a zonated distribution like Dox (but Dox concentration was dozens of Doxol concentrations at the same time), probably due to both zonated characteristics of Dox

accumulation and cytochrome P450 3A (CYP3A) distribution from cHC to pHC[24,25].

It was previously reported that nondegradable gold nanoparticles were eliminated from the body mainly through the hepatobiliary pathway and NPC acted as the barriers for this process[19]. However, for degradable nanoparticles, the situation should be different, as NPC, especially KC, probably acted as the promoter for hepatobiliary elimination of sLip/Dox, considering liposomal Dox accumulation in the liver suddenly decreased when KC was depleted. At 4 h after injection (Fig. 4b), Dox concentration in bile of free Dox treated mice was three times more (*P* < 0.01) than that of sLip/Dox treated mice, but comparable at 12 h and even reversed at 24 h, probably ascribing to quick

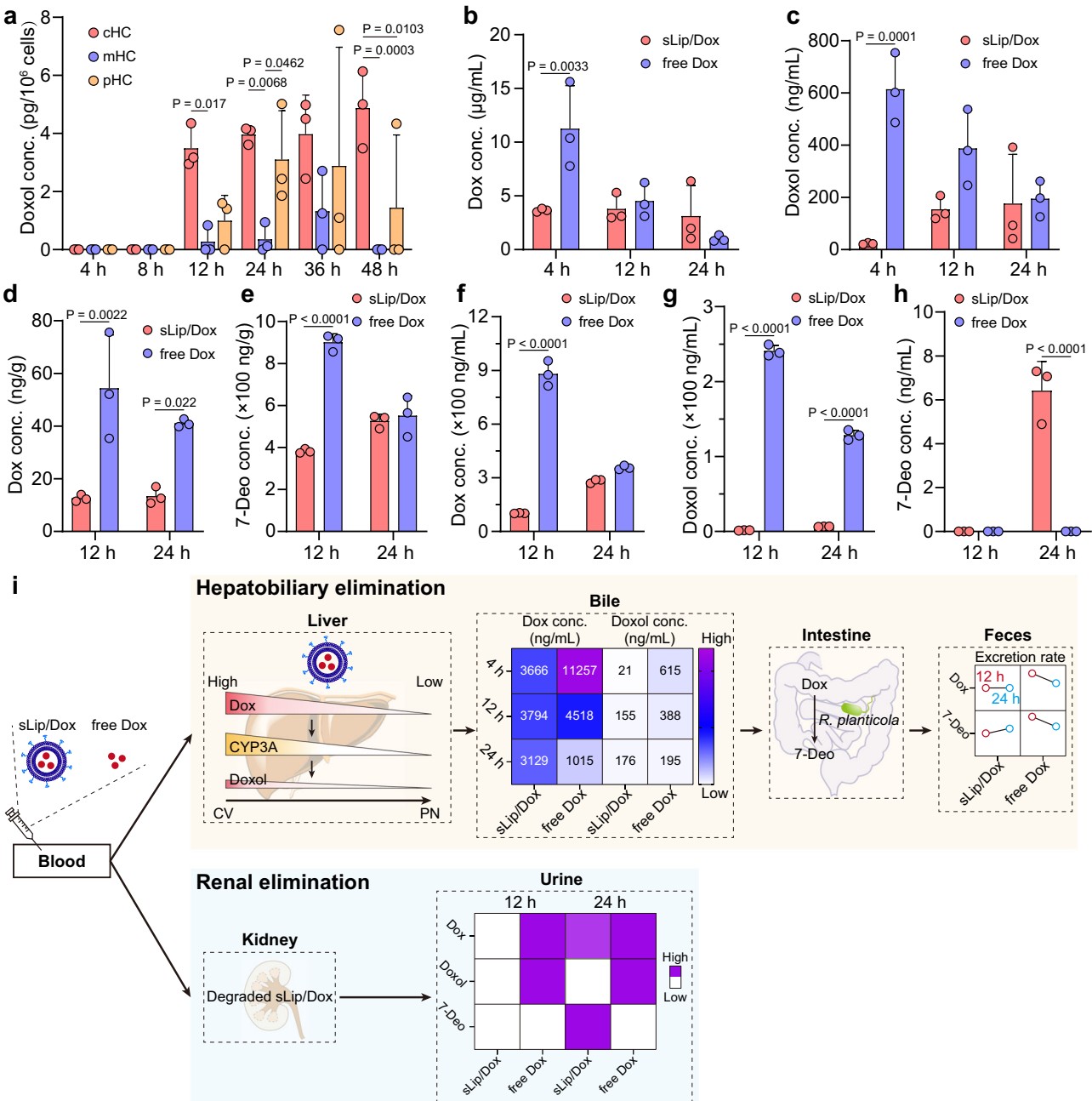

**Fig. 4 | Effect of liposomes encapsulation on metabolism and excretion of Dox.**
**a** Metabolic kinetics of Dox to Doxol in zonated hepatocytes from sLip/Dox treated mice. The bile distribution of Dox (**b**) and Doxol (**c**) at 4, 12, and 24 h post injection. The feces distribution of Dox (**d**) and 7-Deo (**e**) at 12 and 24 h post injection. The urine distribution of Dox (**f**), Doxol (**g**), and 7-Deo (**h**) 12 and 24 h post injection. The sLip/Dox or free Dox was intravenously injected into C57BL/6J mice at a dose of 5 mg/kg Dox. The statistical significance was analyzed by two-way ANOVA multiple comparisons corrected by Sidak's test. *P* values are provided when there are statistical significances (*P* < 0.05). Data were means ± SDs for **a**–**h** (*n* = 3 mice). **i** Injected sLip/Dox or free Dox would excrete through hepatobiliary or renal elimination pathway. For hepatobiliary elimination (liver → bile → intestine → feces), sLip/Dox exhibited a zonated metabolism behavior in the liver as Doxol concentration decreased from cHC to pHC, adapting to both Dox and CYP3A in

zonated distribution. As for excretion into bile, Dox encapsulated in liposomes was slower and constant compared to free Dox, and also was less metabolized before entering bile. In the intestine, Dox would be metabolized into 7-Deo by reductive deglycosylation with the aid of gut microbiome, for example, *R. planticola*. For excretion in feces, compared to free Dox treated mice, sLip/Dox treated mice displayed slower Dox excretion and metabolism to 7-Deo, and from 12 to 24 h, Dox from sLip/Dox excretion into feces and metabolizing into 7-Deo displayed an increasing trend while reversed for free Dox. For renal elimination, sLip/Dox could be degraded and excreted into the urine. Compared to free Dox treated mice, both Dox and two metabolites was much less in the urine from sLip/Dox treated mice except for 24 h, that 7-Deo excretion was higher than the free Dox group. Source data are provided as a Source Data file.

elimination of free Dox while relatively long-acting for liposomal Dox. As for metabolites distribution in bile (Fig. 4c), there was only Doxol detected and displayed 23 times more (*P* < 0.001) in free Dox treated mice than sLip/Dox treated mice at 4 h post injection, and suddenly decreased to 2.5 folds at 12 h and 1.1 folds at 24 h, revealing that quickly

infused free Dox suffered a fast metabolism and secretion into the gall bladder. While for liposomal Dox, metabolism seemed to be slow, as much more Dox was distributed in bile than Doxol at the same time compared to free Dox, especially at 4 h ($C_{Dox}/C_{Doxol}$ = 165.6 for sLip/Dox while 18.3 for free Dox treated mice). Dox in bile may be secreted into

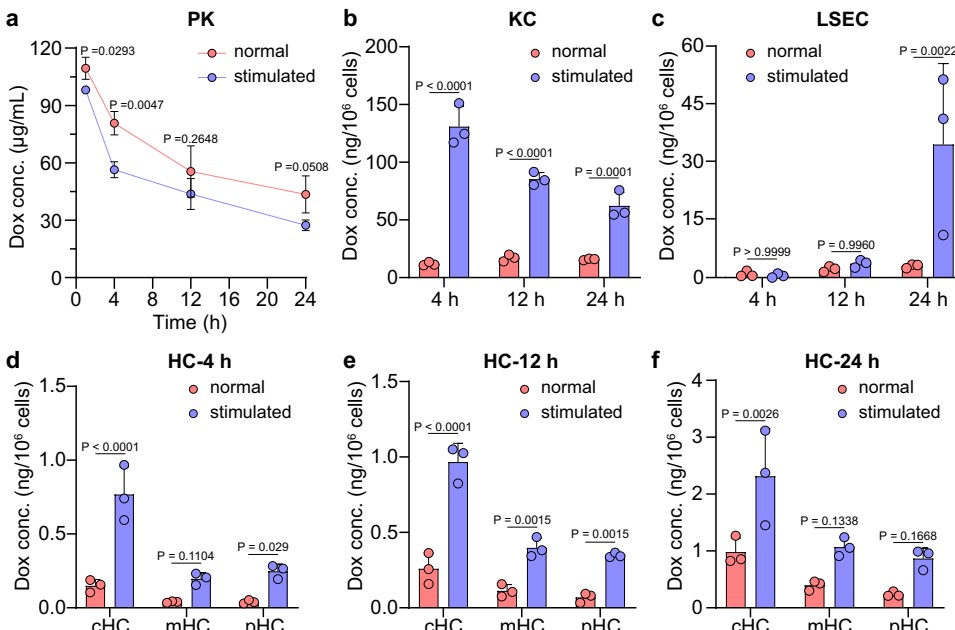

**Fig. 5 | Anti-PEG antibody effects on blood circulation and intrahepatic cellular distribution of sLip/Dox. a** Plasma pharmacokinetics (PK) profiles of Dox in normal and pre-stimulated mice. Dox distribution in KC (**b**), LSEC (**c**), and zonated HC (**d**–**f**) was evaluated. Compared to normal mice, stimulated mice were injected with sLip at a dose of 5 mg/kg HSPC 5 days before being treated by sLip/Dox. Both the normal and pre-stimulated mice were intravenously injected with sLip/Dox at a dose of 5 mg/kg Dox. The statistical significance was analyzed by two-way ANOVA multiple comparisons corrected by Sidak's test except for (**a**) (two-tailed unpaired $t$-test). Data were means ± SDs for **a**–**f** ($n = 3$ mice). Source data are provided as a Source Data file.

the intestine and excreted out of the body in feces. As shown in Fig. 4d, Dox distribution was over three times more ($P < 0.01$) in feces from free Dox-treated mice than that from sLip/Dox treated mice at 12 and 24 h after administration. Besides, Dox concentration in feces from free Dox treated mice indicated a decreased excretion rate from 12 to 24 h, while for sLip/Dox treated mice, increased Dox concentration from 12 to 24 h revealed a constant excretion of Dox into feces. As shown in Fig. 4e, 7-Deo in feces was much more than Dox in both groups ($P < 0.001$ at 12 h), and exhibited an increasing trend in sLip/Dox treated mice from 12 to 24 h, while reversed in free Dox treated mice. Considering Dox accumulated in HC and bile was more than Doxol from 4 to 12 h, there would be limited metabolism for Dox in the liver. As a result, an unexpectedly large amount of 7-Deo in feces was probably due to metabolism in the intestine with the aid of *Raoultella planticola* (*R. planticola*)[26].

Renal excretion is difficult for most nondegradable nanoparticles, as the sizes of reported nondegradable nanoparticles usually surpass the size of a glomerular filter (about 6 nm)[27]. However, the situation could be different for those degradable nanoparticles, which could be degraded and excreted by the renal pathway. As shown in Fig. 4f, for free Dox-treated mice, the Dox concentration in urine was 881.1 and 357.2 ng/mL at 12 and 24 h post injection, about 8.6 ($P < 0.001$) and 1.3 times more than sLip/Dox treated mice, respectively. Besides, there was also more Doxol detected in urine from free Dox treated mice (Fig. 4g), about 158.2 and 19.7 times more ($P < 0.001$) than sLip/Dox treated mice at 12 and 24 h post injection, respectively. As for 7-Deo excretion in urine (Fig. 4h), the concentration was about 6.4 ng/mL in urine from sLip/Dox treated mice at 24 h post injection, but was not detected at 12 h and in the free Dox treated group. Totally sLip/Dox endowed Dox a slower excretion through renal elimination compared to free Dox injection, probably due to the extra degradation and drug release process.

## Anti-PEG antibodies intensify KC capture-HC accumulation pathway
From the 1970s, the PEGylation strategy for biomacromolecules or nanoparticles was harnessed to facilitate various marketed products[28].

However, with the wide use of PEGylated nanomedicine in clinics, there are some problems uncovered, especially anti-PEG antibodies related to accelerated blood clearance (ABC) phenomenon and hypersensitivity reactions[29]. As previously reported, a pre-dose of PEG-HSPC liposomes via intramuscular injection or pre-injection with PEG-based pharmaceutical excipients (poloxamer 188 or poloxamer 407), would induce an obvious lift of anti-PEG antibodies level, eventually boosted blood clearance of second dose of PEGylated liposomal doxorubicin[30,31]. However, where could be the potential destination of the cleared nanoparticles? In our previous work, it was reported that natural IgM, instead of IgG, played an important role in regulating the in vivo performance of liposomes, for example, complement activation effects and blood circulation[32]. In this part, the effects of anti-PEG IgM on sLip/Dox distribution in hepatic cells were evaluated. Compared to normal mice (with anti-PEG IgM at baseline level), the plasma level of anti-PEG IgM in stimulated mice obviously improved 5 days after stimulated by sLip (Supplementary Fig. 14). With the intervention of anti-PEG IgM, a typical ABC phenomenon was exhibited in stimulated mice (Fig. 5a and Supplementary Fig. 15). As in stimulated mice, plasma concentration of Dox was continuously lower than normal mice, and exhibited a significantly less ($P < 0.01$) AUC$_{1-24\,h,\,Dox}$ value from 1 to 24 h after sLip/Dox injection. As for intrahepatic cellular distribution, Dox concentration in KC of stimulated mice significantly increased, about 12 times more at 4 h, five times more at 12 h and four times more ($P < 0.001$) at 24 h than normal mice, respectively (Fig. 5b). While in LSEC, Dox concentration was still low both in normal and stimulated mice at 4 and 12 h, except for 24 h that Dox concentration in LSEC was ten times more ($P < 0.01$) in stimulated mice compared to the normal mice (Fig. 5c), probably due to anti-PEG IgM mediated Fc receptor-dependent capture[33]. Especially with higher Dox concentration in KC, Dox in HC was also significantly increased in stimulated mice from 4 to 24 h post injection, about two to six times more in comparison to the corresponding HC along the lobule porto-central axis in normal mice (Fig. 5d–f), which further testified KC capture-HC accumulation pathway for liposomal Dox transport.

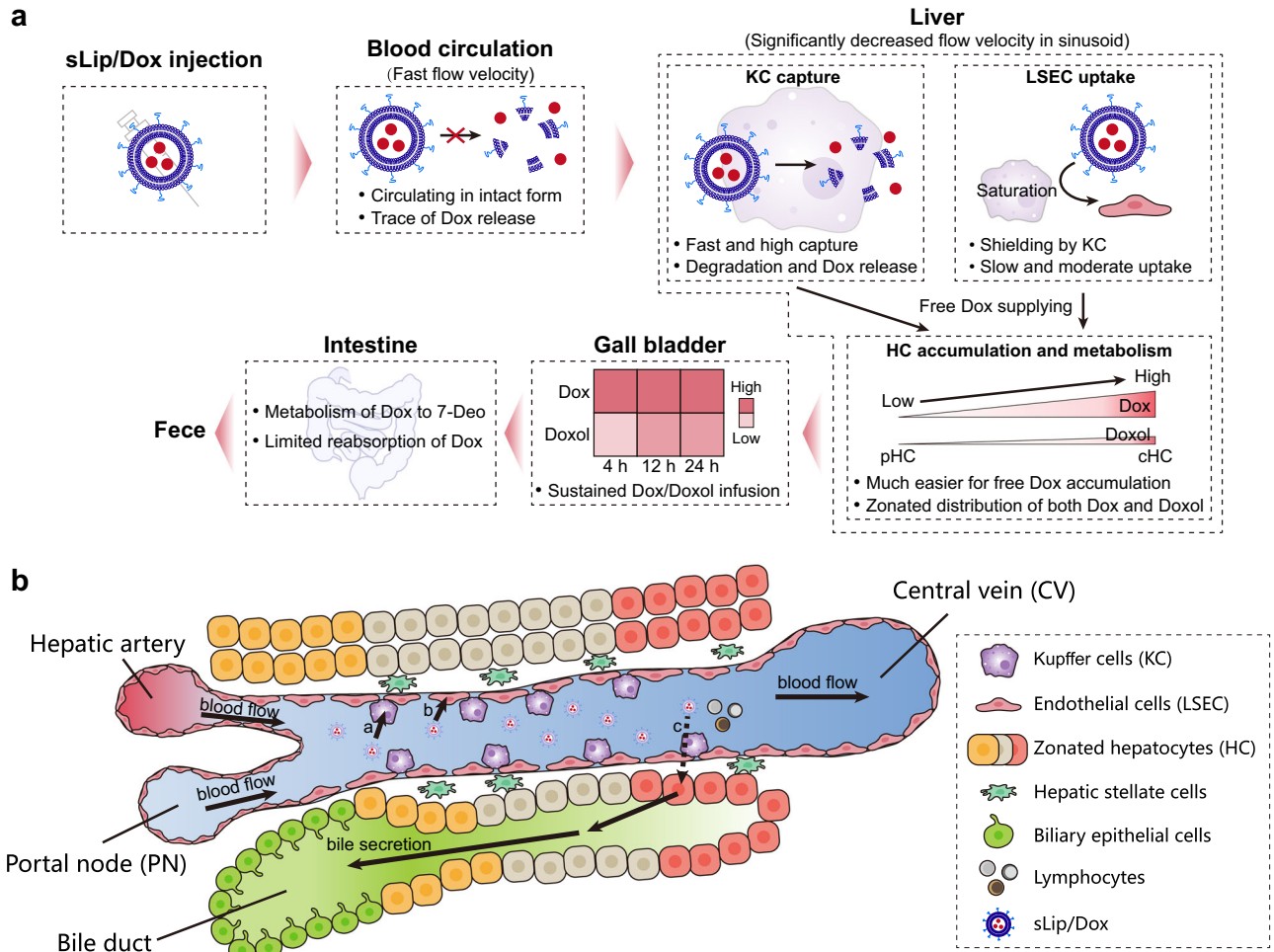

**Fig. 6 | A scheme diagram of intrahepatic cellular interaction of sLip/Dox from blood circulation and following excretion. a** The whole journey of sLip/Dox suffering hepatobiliary elimination. Intravenously injected sLip/Dox is stable in blood circulation and maintains intact to reach liver sinusoid, where sLip/Dox is mainly captured by KC and LSEC. KC tends to display fast and high endocytosis of sLip/Dox to cause the degradation of liposomes and Dox release to extracellular sites. LSEC uptake of sLip/Dox seems to be shielded by KC and captures sLip/Dox in a slow and moderate manner after KC saturation. Dox accumulation and metabolism in HC is supplied from KC/LSEC in a free form, and displays a zonated distribution from cHC to pHC, subsequently Dox and the metabolites (mainly Doxol) constantly infuse into the gall bladder. Dox secreted into the intestine mainly suffers metabolizing to 7-Deo, and would be a limited reabsorption due to P-glycoprotein, eventually out of the body in feces. **b** sLip/Dox infusion in liver sinusoid would be directly captured by KC (**a**), LSEC (**b**), and transferred into HC (**c**), subsequently suffering bile secretion.

## Discussion

Lipid-based nanocarriers promote most nanomedicines in clinics and are experiencing unprecedented progress considering excellent performance during the COVID-19 epidemic[3,34], but still with various problems unsolved. Taking Doxil® as an example, although alleviating cardiotoxicity and improving the tolerant dose of doxorubicin, Doxil® in clinic usually causes an unexpected hand-foot syndrome and displayed limitedly improved overall efficiency compared to the free doxorubicin[35,36]. Besides, for actively targeted lipid-based nanocarriers, there is still no successful clinical translation, although with a development history for several decades[37]. With the widespread vaccination of mRNA vaccines, there are also some autoimmune hepatitis reports[38]. The compromised efficacy and safety of lipid-based nanocarriers could be largely due to the absence of understanding about complicated in vivo fate, which tends to cause uncontrollable interactions with the human body. Considering intrahepatic accumulation occupied most tissue distribution for intravenously injected nanoparticles[17], the detailed intrahepatic fate of PEGylated liposomal doxorubicin was explored in this work.

Totally the journey of sLip/Dox from blood circulation to excretion out of the body, especially the intrahepatic transport/distribution/metabolism/excretion, was speculated as in Fig. 6a. As previously reported[21], more than 98% of sLip/Dox in blood circulation kept intact, leading to it perfusing into liver sinusoid mainly as the nanoparticle form. As for intrahepatic transport, in spatial view, perfused sLip/Dox would firstly interact with NPC in liver sinusoid (Fig. 6b), and KC and LSEC were evaluated as major participants for the capture of sLip/Dox in liver sinusoid. Briefly, KC exhibited a fast and massive capture on sLip/Dox, while LSEC behaved as lagging and moderate on the uptake. Considering free Dox was overlooked by those two cells, liposome encapsulation endowed the payload distribution in KC and LSEC through a receptor-mediated pathway[17,33]. Notably, when KC were depleted, intrahepatic accumulation of sLip/Dox in the liver exhibited a significant decrease, but Dox distribution in LSEC exhibited a significant increase with KC depletion, revealing a possible shielding effect of KC on LSEC endocytosis of liposomes with a small size (90 nm). Previously it was reported that KC/LSEC impeded hepatobiliary elimination of non-biodegradable gold nanoparticles, as more nanoparticles excretion into feces with KC depleted[19]. But for biodegradable sLip/Dox, the situation reversed, as KC (probably accompanied by LSEC) tended to capture and degrade sLip/Dox to release Dox, which subsequently entered HC through fenestration and elimination.

Adapt to the physiological function, HC displayed a special zonated characteristic in metabolism functions and substance distribution[39]. Although with various researches focused on zonation characteristics of HC, there was still no research about the effects on the intrahepatic distribution of nanoparticles. For sLip/Dox, zonated distribution was not obvious 1 h post injection, probably due to most intrahepatic sLip/Dox was still entrapped in KC. But after 4 h post injection, which was an enough time period for sLip/Dox degradation in KC[19], the released liposomal Dox was quickly accumulated in HC and displayed a zonated distribution from cHC to pHC. A large amount of intrahepatic accumulation of Dox from sLip/Dox led to not only a reduced distribution in extrahepatic lesion sites, but the burden for liver metabolism and potential cytotoxicity. Although more than eight metabolite structure has been identified[40], it was found that Doxol and 7-Deo were the main metabolites of Dox suffering hepatobiliary elimination. Considering Doxol was still pharmacologically active (7-Deo was not) as doxorubicin[41], which could be toxic to HC and other cells in the excretion pathway, the metabolic profile of Doxol in zonated HC was explored and also displayed a zonated distribution, probably ascribing to both decreasing trend of Dox and CYP3A from cHC to pHC[42].

Summarized elimination pathway of sLip/Dox as in Fig. 4i, along the hepatobiliary elimination, gall bladder was the important transfer station receiving cargos from HC, following secretion into intestine with bile, and eventually eliminated in feces. Dox concentration in bile was much higher than Doxol, indicating that intrahepatic Dox left the liver mainly in an unchanged structure. Besides, compared to free Dox, sLip/Dox displayed an extended Dox distribution in the gall bladder, probably derived from long circulation of sLip/Dox and slower infusion of encapsulated Dox in HC due to extra disposition process by KC. In intestine, Dox would suffer a metabolism to 7-Deo with the aid of *R. planticola*, leading to enhanced distribution of 7-Deo in feces in comparison to Dox. Besides, there would be limited reabsorption of Dox in the intestine due to P-glycoprotein[43].

The results in the present study revealed that regulation of KC capture could be efficient to control in vivo behaviors of liposomes. For example, to control liposomes in smaller sizes, or inhibit complement activation, which could lower KC capture of liposomes[44], would benefit to avoid excessive intrahepatic drug accumulation, meanwhile reduce potential intrahepatic metabolism burden and even toxicity to cells along the hepatobiliary elimination pathway. Besides, for precision medication of PEGylated liposomal doxorubicin in a clinic, as lifted anti-PEG antibodies level was widely detected[45] and would eventually upregulate doxorubicin accumulation in hepatocytes and following excretion into the intestine, strictly controlled dosage should be necessary for patients with higher pre-existing anti-PEG antibodies to avoid probable toxicity of doxorubicin to liver or intestine.

From first reported in the 1960s to nowadays, lipid-based nanocarriers are continuously promising for drug delivery and promote most nanomedicines in clinics. However, there are still some problems unsolved, containing attenuated efficacy and unexpected side effects, largely due to missing information about the in vivo process. Considering liver accumulation dominates in vivo distribution, intrahepatic transport/metabolism/excretion of PEGylated liposomal doxorubicin were explored in this work to promote understanding of nanomedicine-body interaction. It was indicated that intrahepatic KC, which was reported with powerful capture on nanoparticles, dominated following an intrahepatic process of PEGylated liposomal doxorubicin. Briefly, sLip/Dox captured by KC would be degraded to extracellularly release Dox, which subsequently entered HC and exhibited a zonated distribution in HC along the lobule porto-central axis. In this process, the amount of liposomal Dox trapped in KC was positively correlated to following Dox transport into HC, and KC was also indispensable for zonated drug distribution in HC. In short, this work provided precautions for further development or clinical use of nanomedicines, as KC capture should be well-regulated to achieve targeted drug delivery, whether for intrahepatic or extrahepatic diseases.

## Methods

### Reagents and antibodies

Hydrogenated soy phosphatidylcholine (HSPC), cholesterol (Chol), *N*-(carbonyl-methoxy polyethylene glycol$_{2000}$)-1, 2-distearoyl-*sn*-glycerol-3-phosphoethanolamine (DSPE-PEG$_{2000}$), and 1, 2-distearoyl-*sn*-glycero-3-phosphoethanolamine-*N*-[(polyethylene glycol)-2000]-amine (DSPE-PEG$_{2000}$-NH$_2$) were purchased from A.V.T. Pharmaceutical (Shanghai, China). Sephadex® G-50 were purchased from Sigma (St. Louis, MO, USA). Doxorubicin hydrochloride (Dox), daunorubicin hydrochloride (Dau), and 5-carboxyfluorescein *N*-succinimidyl ester (FAM-NHS) were purchased from Macklin Biochemical (Shanghai, China). Metabolites of doxorubicin (Supplementary Fig. 16), doxorubicinol hydrochloride (Doxol), and 7-deoxy doxorubicin aglycone (7-Deo) were purchased from Toronto Research Chemicals (Toronto, Canada). Collagenase IV was purchased from MKbio (Shanghai, China). Percoll was purchased from GE Healthcare (Pittsburgh, USA). HRP-labeled goat anti-mouse IgM mu chain (Cat#AB97230, Lot#1028601-18) was purchased from Abcam (Cambridge, USA). The 3, 3′, 5, 5′ -tetramethylbenzidine (TMB) chromogen solution was acquired from Beyotime Biotechnology (Nantong, China). The other antibodies used in this work are listed in Table 1. The other reagents were of marketed analytical grade.

### Animals

Healthy male C57BL/6J mice (7–8 weeks) were obtained from the Experimental Animal Center of Fudan University, and maintained at $22 \pm 2\,°C$ and $50 \pm 5\%$ humidity on a 12 h light−dark cycle with access to standard rodent chow (Cat number: 1010088, Jiangsu Xietong Pharmaceutical Bio-engineering Co., Ltd.) and water ad libitum. All the animals were treated according to protocols that were approved by the Ethical Committee of Fudan University (Approved number: 20230301-100 and 20210302-022). All the mice used in this work were implemented general anesthesia by intraperitoneal injection of pentobarbital sodium (50 mg/kg), and euthanized by an overdose of intraperitoneally injected pentobarbital sodium (150 mg/kg). Apart from healthy male C57BL/6J mice used in this work, the other two animal models were further established. For KC-depletion mice, the healthy male C57BL/6J mice were intravenously injected with 200 μL clodronate liposomes (from Vrije Universiteit Amsterdam), and after 2 days the liver nonparenchymal cells were isolated and confirmed KC-depletion status by flow cytometry (Fig. 1d). For anti-PEG antibody pre-existing mice, the healthy male C57BL/6J mice were intravenously injected with sLip at a dose of 5 mg/kg HSPC, and after 5 days anti-PEG IgM antibody level in serum was detected by an ELISA method[21]. Briefly, a medium-binding 96-well plate was coated with DSPE-PEG$_{2000}$ (2 μg/well) in ethanol overnight. After blocking with 5% BSA in 10 mM PBS at 37 °C for 1 h and washed with 0.1% Tween-PBS (PBST), 100 μL mice plasma with serial dilutions by 0.1% BSA-PBS was added to the plate and incubated for 1 h at 37 °C. After washing with PBST, diluted HRP-labeled anti-mice IgM antibody was added with further 1-h incubation at 37 °C. Then TMB was added for 8 min chromogenic reaction and the reaction was terminated by 0.18 M sulfuric acid. The OD$_{450\,nm}$ was measured by a microplate reader. The established models were used within 2 to 4 days after injection of clodronate liposomes for KC-depletion mice, and within 5 to 7 days after injection of sLip for anti-PEG antibody pre-existing mice.

### Preparation and characterization of liposomes

The composition of PEGylated liposomes (sLip) used in this work is referred to Doxil®, which was composed of HSPC, Chol, and DSPE-PEG$_{2000}$ at a mass ratio of 9.58: 3.19: 3.19, and non-PEGylated

**Table 1 | The list of antibodies used in this work**

| Antibodies | Company | Cat# | Lot# | Clone | Application |
|---|---|---|---|---|---|
| BV421 anti-mouse F4/80 | Biolegend | 123137 | B422926 | BM8 | KC marker |
| APC anti-mouse CD146 | Biolegend | 134712 | B396861 | ME-9F1 | LSEC marker |
| BV421 anti-mouse CD19 | Biolegend | 115538 | B399079 | 6D5 | hBC marker |
| APC anti-mouse CD3 | Biolegend | 100236 | B413435 | 17A2 | hTC marker |
| BV421 anti-mouse CD324 | Biolegend | 147319 | B324426 | DECMA-1 | pHC marker |
| APC anti-mouse CD73 | Biolegend | 127210 | B291453 | TY/11.8 | cHC marker |
| BV421 anti-rat IgG2a | Biolegend | 400535 | B340002 | RTK2758 | Isotype control |
| APC anti-rat IgG2a | Biolegend | 400511 | B336072 | RTK2758 | Isotype control |
| APC anti-rat IgG2b | Biolegend | 400611 | B314628 | RTK4530 | Isotype control |
| BV421 anti-rat IgG1 | Biolegend | 400429 | B343563 | RTK2071 | Isotype control |
| APC anti-rat IgG1 | Biolegend | 400411 | B337839 | RTK2071 | Isotype control |
| FcR Blocking Reagent | Miltenyi | 130-092-575 | 5240407396 | - | FcR blocking |

liposomes (Lip) were prepared without DSPE-PEG$_{2000}$. To prepare the FAM-labeled PEGylated liposomes (FAM-sLip), one-tenth of DSPE-PEG$_{2000}$ was substituted by DSPE-PEG$_{2000}$-FAM (Supplementary Fig. 17). After the lipids ethanol solution rotary evaporated at 60 °C and dried in vacuum for overnight, the obtained thin film was hydrated with normal saline at 60 °C. The dispersion was then extruded through polycarbonate membranes with pore sizes of 400, 200, 80, and 50 nm at 60 °C, followed by the separation of plain lipids and assembled liposomes by Sephadex G-50 column at room temperature to obtain the FAM-sLip. For preparation of liposomal Dox, the lipid thin film was hydrated with 320 mM ammonium sulfate (added 1 mL per 9.58 mg HSPC) at 60 °C, followed by extrusion as in FAM-sLip preparation (the dispersion was extruded through only 400 nm membranes for 300 nm liposomes, while successively extruded through 400, 200, and 100 nm membranes for 120 nm liposomes). After ammonium sulfate in the external liquid phase was replaced by normal saline in the G-50 column, Dox was incubated with liposomes at the ratio of 10% w/w (Dox/HSPC) at 60 °C for 20 min, followed by separation of Dox and liposomal Dox by G-50 column at room temperature to obtain sLip/Dox or Lip/Dox. Prepared liposomes were stored at 4 °C before using.

The morphology of liposomal drug was observed under a Tecnai G2 F20 cryogenic transmission electron microscopy (cryo-TEM, FEI, Holland). Particle sizes, polydispersity index (PDI), and zeta potentials of liposomal drugs were tested using the Zetasizer Nano ZS90 (Malvern Panalytical, UK). Lipids concentration was evaluated using an ammonium molybdate colorimetric assay. Briefly, the liposomes were incubated with 10 M sulfuric acid at 175 °C for 1 h, following incubation with hydrogen peroxide at 175 °C for 1 h. Then the cooled solution was incubated with ammonium molybdate and ascorbic acid at 100 °C for 8 min. The optical density of the solution at 812 nm was detected by the microplate reader (Tecan, Switzerland) and the HSPC concentration was calculated using sodium dihydrogen phosphate as standard solutions. Dox concentration was detected by HPLC (Agilent, USA) on a Diamonsil 3-μm C18(2) column (100 × 2.1 mm). The mobile phase was 40% acetonitrile solution. The peak area of Dox was quantified at room temperature under the UV detector at 254 nm.

**Pharmacokinetics**

Healthy male C57BL/6J mice (7–8 weeks) were injected with FAM-sLip at a dose of 50 mg/kg HSPC or liposomal Dox/free Dox at a dose of 5 mg/kg Dox via the tail vein. After intraperitoneal injection of pentobarbital sodium (50 mg/kg) for general anesthesia, blood was collected from the anterior canthi and maintained at 4 °C. For pharmacokinetics evaluation of FAM-sLip (three mice for each timepoint), the blood was centrifuged (3500×$g$, 5 min, 4 °C) to separate the plasma. Then 30 μL

plasma gathered at different timepoints from different injected mice was mixed with 90 μL normal saline and the fluorescence intensity was detected by the microreader at the excitation wavelength of 490 nm and the emission wavelength of 520 nm (Tecan Spark, Switzerland), using 30 μL blank plasma from untreated mice mixing with FAM-sLip in 90 μL normal saline in gradient concentrations to establish a standard curve. For pharmacokinetics evaluation of sLip/Dox (three mice for each timepoint), plasma was gathered as above, and Dox was extracted by chloroform/methanol for further HPLC detection. Briefly, 10 μL plasma, 30 μL 10 mM PBS, 60 μL distilled water, 100 μL Dau methanol solution (10 μg/mL), and 400 μL chloroform were mixed and vortexed for 2 min, then the mixture was centrifuged (6000 × $g$, 15 min, 4 °C) and the bottom layer was extracted for drying overnight. The extractive was re-dissolved in 120 μL 40% acetonitrile solution and detected by HPLC. Besides, for pharmacokinetics evaluation of injected free Dox (three mice for each timepoint), due to extremely low concentration even after 1 h, Dox concentration was detected by a liquid chromatography-tandem mass spectrometry (LC-MS/MS) method. Briefly, 10 μL diluted plasma (pre-diluted by 10 mM PBS for 10 folds), 90 μL 10 mM PBS, 20 μL Dau methanol solution (50 ng/mL), 80 μL methanol, and 400 μL chloroform were mixed and vortexed for 2 min, following the same procedures as in sample handling for plasma from liposomal Dox treated mice. The method of LC-MS/MS detection is shown in the next part.

**LC-MS/MS detection**

Dox and its major metabolites (Doxol and 7-Deo) at relatively low concentration was detected by a LC-MS/MS method[40]. Briefly, those compounds were separated on a Diamonsil C18(2) column (100 mm × 2.1 mm, 3 μm) at room temperature using a UPLC Nexera coupled with an LC-MS-8060 mass spectrometer (Shimadzu, Japan) by gradient elution of ultrapure water (moving phase A, contained 0.1% formic acid) and acetonitrile (moving phase B, contained 0.1% formic acid) at a flow rate of 0.3 mL/min as following: 20% B for 1 min, 20 to 30% B for 1 min, 30 to 70% B for 2 min, 70 to 90% B for 2 min, 90% B for 2 min, 90 to 20% B for 0.1 min, 20% B for 1.9 min. The injection volume was 10 μL and the samples were sealed in an autosampler tray at 4 °C to inhibit evaporation or degradation. The electrospray source settings for ionization in positive mode were as follows: nebulizing gas flow, 3 L/min; drying gas flow, 10 L/min; heating gas flow, 10 L/min; interface voltage, 4 kV; interface current, 0.6 μA; interface temperature, 300 °C; DL temperature, 250 °C; heat block temperature, 400 °C; conversion dynode, 10 kV; detector voltage, 2.36 kV; CID gas, 270 kPa. The collision energies and multiple reaction monitoring transitions for quantification were listed in Table 2. Data processing was executed using the Postrun module in LabSolutions LC-MS software version 5.6.

**Table 2 | Multiple reaction monitoring transitions, collision energies (CE), and quantitative range (QR) for identification and quantification of doxorubicin and its metabolites**

| Compound | Precursor (m/z) | Product (m/z) | CE (eV) | QR (ng/mL) |
|---|---|---|---|---|
| Dox | 544.30 | 397.25 | −14 | 0.5–200 |
| Doxol | 546.30 | 399.25 | −15 | 0.1–200 |
| 7-Deo | 399.35 | 381.25 | −20 | 0.5–50 |
| Dau | 528.30 | 321.10 | −25 | Interior standard |

## Organ distribution

To explore Dox accumulation in different organs, mice (three mice in each group at different timepoints) were euthanized with an overdose of intraperitoneally injected pentobarbital sodium (150 mg/kg), and the heart, liver, spleen, lung, and kidneys were dissected, weighted and homogenized with tenfold volume of 5% Triton-X 100 in 10 mM PBS at 4 °C after infusion with 20 mL normal saline. For LC-MS/MS detection, 200 µL homogenate, 20 µL Dau methanol solution (50 ng/mL), 80 µL methanol, and 400 µL chloroform were mixed and vortexed for 2 min, then the mixture was centrifuged ($6000 \times g$, 15 min, 4 °C) and the bottom layer was extracted for drying overnight. The extractive was re-dissolved in 120 µL 40% acetonitrile solution, and the Dox concentration was detected by LC-MS/MS.

## Liver cells isolation

Liver cells were isolated combining a two-step perfusion digestion and flow cytometry sorting procedure. Briefly, in general anesthesia by pentobarbital sodium (50 mg/kg), mice were put back on the dissection board, and the peritoneum was dissected to uncover the central vein and portal vein, following constant perfusion with 37 °C isotonic HBSS-0.5 mM EGTA buffer (20 mL) and isotonic HBSS-3 mM $CaCl_2$-100 U/mL collagenase IV buffer (10 mL) from central vein (inlet) to portal vein (outlet). The intact liver was then transferred into a 60 mm Petri dish containing 3 mL collagenase buffer, bluntly dissected into small pieces about $1 mm^3$, and digested at 37 °C for 15 min. Then, the mixture was filtered through a 70-µm cell strainer and washed with cold isotonic HBSS-3 mM $CaCl_2$ buffer (20 mL). The total collected cells suspension was centrifuged ($50 \times g$, 4 min, 4 °C) to separate total NPC in supernatant while HC in the precipitate. For NPC purification, the supernatant was centrifuged ($700 \times g$, 5 min, 4 °C), resuspended in 1 mL 10 mM PBS and gently put on a 25%/50% (2.5 mL/2 mL) Percoll gradient for centrifugation ($1800 \times g$, 15 min, 4 °C). The total Percoll layer was then poured into fivefold volume 10 mM PBS and centrifuged ($700 \times g$, 5 min, 4 °C) to gather the precipitate (containing KC, LSEC, hBC, and hTC). Cells were further isolated by flow cytometry sorting (BD, USA) based on differences in surface marker expression. The HC in the precipitate were washed by cold isotonic HBSS-3 mM $CaCl_2$ buffer three times, and the precipitate was gathered as total HC. To separate HC from different lobule zones, total HC was resuspended in 2 mL 10 mM PBS and gently put on a 30%/42%/52%/70% (2.5 mL: 2 mL: 1.5 mL: 1 mL) Percoll gradient for centrifugation ($400 \times g$, 15 min, 4 °C), and the cells at interfaces between different Percoll layers were poured into fivefold volume 10 mM PBS and centrifuged ($400 \times g$, 3 min, 4 °C) to gather the precipitate as HC from different lobule zones (cHC, from interface between 30 and 42% Percoll layers; mHC, from interface between 42 and 52% Percoll layers; pHC, from interface between 52 and 70% Percoll layers).

## Intracellular distribution of FAM-sLip and sLip/Dox

Healthy male C57BL/6J mice (7–8 weeks) were injected with FAM-sLip at a dose of 50 mg/kg HSPC via the tail vein. Three mice were used at each timepoint for the experimental group, while three untreated mice were used as the negative control. After blocking by FcR Blocking Reagent (FcR, at a dilution of 1: 10), isolated NPC were divided into two

parts: one part was stained with BV421 anti-mouse F4/80 and APC anti-mouse CD146 antibodies to analyze FAM-sLip distribution in KC and LSEC; the other part was stained with BV421 anti-mouse CD19 and APC anti-mouse CD3 antibodies to analyze FAM-sLip distribution in hBC and hTC. For zonated HC obtained by Percoll gradient centrifugation, after stained with FcR (at a dilution of 1: 20), pHC was stained with BV421 anti-mouse CD324 antibody while cHC was stained with APC anti-mouse CD73 antibody. Cells from untreated mice liver was used as the blank control after incubation with isotype control antibodies. All the antibodies were used at a dilution of 1: 100. FAM-sLip distribution in various cells was detected by flow cytometry (Agilent, USA).

To explore the intracellular distribution of liposomal Dox or free Dox, mice were intravenously injected with sLip/Dox or free Dox at a dose of 5 mg/kg Dox (three mice at each timepoint for each group). At different timepoints, intrahepatic KC and LSEC were isolated by flow cytometry sorting with cells number recorded (Supplementary Fig. 18), and cells resuspended in 10 mM PBS were mixed with 20 µL Dau methanol solution (50 ng/mL), 80 µL methanol, and 400 µL chloroform, and prepared as above. For HC from different lobule zones, cells were resuspended in 200 µL 10 mM PBS. Then, the HC was mixed with 20 µL Dau methanol solution (50 ng/mL), 80 µL methanol, and 400 µL chloroform, and prepared as above. Intracellular concentration of Dox and its metabolites was detected by LC-MS/MS.

## Distribution of sLip/Dox in bile, feces, and urine

After liver perfusion, the gall bladder of mice was dissected and the bile was extracted for further detection. The treated mice were also fed in metabolism cages to gather the feces and urine (three mice at each timepoint for each group). To extract the drug in bile, a volume of 1 µL bile was mixed with 99 µL 10 mM PBS, 20 µL Dau methanol solution (50 ng/mL), 80 µL methanol, and 400 µL chloroform, and prepared as above. The feces were weighted and homogenized with tenfold volume of 5% Triton-X 100 in 10 mM PBS at 4 °C, and 100 µL homogenate was mixed with 20 µL Dau methanol solution (50 ng/mL), 80 µL methanol, and 400 µL chloroform, and prepared as above. A volume of 10 µL urine was mixed with 90 µL 10 mM PBS, 20 µL Dau methanol solution (50 ng/mL), 80 µL methanol, and 400 µL chloroform, and prepared as above. The concentration of Dox and its metabolites was detected by LC-MS/MS after drug extraction.

## Statistical analysis

Quantitative data were shown as means ± SDs. Statistical significances of the data were analyzed by the $t$-test or ANOVA multiple comparison using GraphPad Prism 8 software (more detailed information is provided in the figure legend). Considered $^{ns}P > 0.05$ as no significant, $*P < 0.05$ as significant, and $**P < 0.01$ even $***P < 0.001$ as highly significant.

## Reporting summary

Further information on research design is available in the Nature Portfolio Reporting Summary linked to this article.

## Data availability

Data supporting the findings of this study are provided in the main manuscript/Supplementary Information/Source Data file. Source data are provided with this paper.

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

## Acknowledgements

This work was supported by the National Natural Science Foundation of China (82125035 and 32330058 to C.Z., 82273866 to J.Q., and 32101131 to K.J.) and Shanghai Education Commission Major Project (2021-01-07-00-07-E00081 to C.Z.). Thanks to the National Center for Protein Science Shanghai for the Cryo-TEM.

## Author contributions

K.J. contributed to experiments design, LC-MS/MS methods establishment, and manuscript preparation; K.T. contributed to liposomes preparation, characterization, and sample detection; Y.Y. and E.W. contributed to animal models establishment and liver cells isolation; M.Y. and F.P. contributed to pharmacokinetics evaluations and data collection; J.Q. contributed to manuscript revision; C.Z. assumed this research, and contributed to the whole work direction and manuscript revision.

## Competing interests

The authors declare no competing interests.
