## [Peer Review File · Nature Communications]

Reviewers' comments:

Reviewer #1 (Remarks to the Author):

In light of concerns over the efficacy and safety of lipid-based nanomedicines, Jiang et al characterize the intrahepatic traffic and clearance of PEGylated liposomal doxorubicin (sLip/Dox), a lipid-based nanoparticle treatment for cancer, in a murine model. Based on their results, Jiang et al propose a mechanism in which Kupffer cells cause intrahepatic accumulation of sLip/Dox by engulfing the nanoparticles. The Kupffer cells then release free doxorubicin which is taken up by hepatocytes in zone-dependent manner. Finally, the authors find that liposome encapsulation slows excretion of doxorubicin compared to free doxorubicin administration.

Major Comments

1. The authors should discuss their results in the context of previous publications, such as PMID: 37256791 and PMID: 37717659 and identify novel parts of their study that are not previously described in the literature.

2. On line 356 of the manuscript, the authors claim that previous studies show limited efficiency of sLip/Dox in comparison to free doxorubicin. However, the stated reference (O'Brien et al, Ann Oncol, 2004) concludes that "[sLip/Dox] provides comparable efficacy to doxorubicin."

3. Many of the authors' conclusions are substantiated by repeating experiments in KC-depleted mice as they reveal the critical role of KC. As such, the authors should show the abundance of KC before and after KC depletion.

4. Is there a redistribution of lipid-based nanoparticles from Kupffer cells to LSEC? What is the consequence/effect of the of lipid-based nanoparticle uptake by liver sinusoidal endothelial cells (see 12h time point)? Is the uptake by LSECs zonation dependent? Since Kupffer cells concentrate in the periportal regions, is there more uptake by LSEC closer to the central veins?

5. Figure 4: the authors show metabolite / bile acid data. What is the effect of the gut microbiota?

6. In the Significances section, the authors assert liposome encapsulation slows doxorubicin metabolism and excretion. The authors should bolster this point by doing a statistically comparison

of doxorubicin concentrations in sLip/Dox treated mice at the 12 h and 24 h timepoints in Figures 4D, Figure 4E, and Figure 4F.

7. The authors found that doxorubicin concentration in KC increased after stimulating mice with sLip. The authors postulate that this finding could be caused by anti-PEG antibodies mediated Fc receptor dependent capture. It is important to show the titers of anti-PEG IgM and IgG at different timepoints could give more context to the findings in Figure 5.

8. The discussion is overly long and repetitive. A description of the detection of PEG by ELISA should be included in the Experimental Section.

9. On line 26 of the Abstract, the authors report that free Dox “quickly distributed into HC without zoned characteristics.” However, Figure 2C does show some differences by liver zonation.

10. Line 451: How can a lower plasma level of anti-PEG antibodies be achieved in patients?

Reviewer #2 (Remarks to the Author):

Jiang et al has reported the role of Kupffer cell (KC) in intrahepatic accumulation of drug through PEGylated liposomal doxorubicin (sLip/Dox). The report has shown KC dominated intrahepatic capture of sLip/Dox and released the intracellular payload Dox to facilitate zoned distribution in hepatocytes (HC). The report also showed a size dependent KC capture ability of sLip/Dox by increasing the size of liposomes size. The report also found the similar effect in the presence of anti-PEG antibodies. However, intravenously injected free Dox failed to do. The study is systematic, but the results are very obvious as shown by others before.

Major:

1. The novelty of this report is completely missing. This falls in the ‘yet another’ category with liposomal interaction with KC. Kuffer cells are very well known for up taking liposomes. In 1980’s, Roerdink et al showed in their seminal paper ‘Interaction of liposomes with hepatocytes and Kupffer cells’. More recently, as reported by Rooijen et al ‘Kupffer cell depletion by liposome-

delivered drugs: Comparative activity of intracellular clodronate, propamidine, and ethylenediaminetetraacetic acid' (<https://doi.org/10.1002/hep.510230544>). In fact, it is becoming a strategy that Tavares et al, showed the 'Effect of removing Kupffer cells on nanoparticle tumor delivery' due to the same reason (<https://doi.org/10.1073/pnas.1713390114>). The authors, therefore, need to clearly redefine the novelty aspect of the present report.

2. It is obvious therefore that sLip/Dox will interact with Kupffer cells and will increase the hepatic cell delivery as shown in Figure 1B-E and with depletion in Figure 1C-D. Is there anything else that the authors were expected here? If so, it is partly clear.

3. The size dependency also is expected due to the phagocytosis cut-off size. It would have been interesting to see if there are any lower and upper cut-off limits exists for the KC uptake.

4. The other concern is about the reason behind using the PEG molecule. This might well be to increase the circulation time. It would have been interesting to see on how the bare non-stealth version behave. Similarly, it is not clear about the role of anti-PEG antibody. If the authors were expected otherwise, then the hypothesis for this experiment is unclear at this stage.

5. It would have been interesting to see whether the surface charge has any role to play for Kupffer cell up taking. Surface zeta potential of the sLip/Dox might influence the uptake.

6. The authors must show some direct evidence on the extent of drug loading capacity of the proposed system.

7. More proper justification of the exact application of this study is needed, as it is not clearly mentioned.

Reviewer #3 (Remarks to the Author):

This work investigated the fate of liposomes and drugs in the liver and proposed a novel mechanism for the liver processing of liposomes. This is indeed an important topic, which can help the future design of safer and more effective liposomal delivery of drugs. However, upon initial review, some technical and experimental design flaws need to be addressed. More work is needed to support the proposed model of liposome capture and drug metabolism by KC and LSEC. My specific comments are as follows:

1) It is intriguing that with KC depletion, the liver concentration of Dox reduced drastically but concentration in other organs did not change much. Have the authors examined the blood half-life of Dox? Where did the particles go?

2) The authors used $\text{ng}/10^6$ cells as the concentration of drug to speculate the mechanism of drug distribution. However, the absolute number of cells in each liver should also be considered. It is possible that one cell population is very abundant, so even when the concentration calculated as $\text{ng}/10^6$ cells is low, its overall uptake might be more dominant.

3) One major problem of methodology is that fluorescent signal was used to quantify cellular uptake of FAM-sLip in figure 1 and mass spec concentration of Dox was used to quantify uptake of sLip/Dox. The difference of kinetics of the two sLip could be due to the sensitivity of two different testing methods. I recommend the authors to be consistent with quantification methodology, for example, also conjugate FAM to Doxil.

4) More data is needed to support the hypothesis that Dox was released from liposomes in KC and infuse into HC. The decrease concentration of Dox in KC can be achieved without liposomal release. It is also possible that Dox has been degraded or Lip/Dox was released in whole.

5) The majority of the results are bar graphs of drug concentration. I recommend the authors to add some immune fluorescence images of different liver cell populations and drug distribution at different time point.

6) Doxil by itself is toxic and can affect function of liver cells. Please add a control experiment to exclude the effects of cytotoxicity of Doxil.

7) KC and LSEC share several functions and gene expression signatures. Recent research also identified there might be subsets of KC exist that assemble the features of LSEC (doi.org/10.1016/j.cell.2021.12.018). This work distinguishes KC and LSEC with just F4/80 and CD146, which might be over-simplified. More specific characterization of these two populations is highly recommended.

Reviewer #1 (Remarks to the Author):

In light of concerns over the efficacy and safety of lipid-based nanomedicines, Jiang et al characterize the intrahepatic traffic and clearance of PEGylated liposomal doxorubicin (sLip/Dox), a lipid-based nanoparticle treatment for cancer, in a murine model. Based on their results, Jiang et al propose a mechanism in which Kupffer cells cause intrahepatic accumulation of sLip/Dox by engulfing the nanoparticles. The Kupffer cells then release free doxorubicin which is taken up by hepatocytes in zone-dependent manner. Finally, the authors find that liposome encapsulation slows excretion of doxorubicin compared to free doxorubicin administration.

Major Comments

1. The authors should discuss their results in the context of previous publications, such as PMID: 37256791 and PMID: 37717659 and identify novel parts of their study that are not previously described in the literature.

Response: We are grateful to the reviewer for suggestions to improve our work. In the two published papers as the reviewer mentioned (PMID: 37256791 and PMID: 37717659), the authors investigated production of anti-PEG antibodies and their effects on blood clearance of PEGylated nanoparticles (those particles were finally captured by mononuclear phagocyte system/MPS in the liver). Such kinds of studies have been published by many investigators, including our group. However, there were no systematic study on the whole process of intrahepatic transport, distribution, metabolism and excretion of liposomes, which would be crucial for guiding the development of liposome-based therapeutics for both intrahepatic and extrahepatic diseases. However, the intrahepatic fate of liposomal formulations (including transport from blood vessels to hepatic cells, intercellular transport and distribution, drug release and metabolism, and excretion) remains elusive, which are crucial for development of new formulations and clinical application of the approved ones.

In our previous version of the manuscript, we mainly focused on **the intrahepatic transport process of PEGylated liposomal doxorubicin** (which is the first FDA-approved liposomal anti-cancer therapeutics), especially on **quantitative contribution of various hepatic cells to the intrahepatic transport process**. In addition, the effects of physicochemical properties of liposomes (sizes, surface modifications, etc.) and physiological environment (Kupffer cells, anti-PEG IgM, etc.) on such process were also illustrated. These points have not been reported before. Kupffer cells have been well known for massive capture of liposomes. While to the best of our knowledge, we are the first to reveal the **pivotal effects of Kupffer cells on intrahepatic transport of liposomes, not only massively took up liposomes, but also determined *in vivo* transport, distribution, metabolism and excretion of the payload (such as the loaded doxorubicin)**.

To make the description clearer, we modified our manuscript in **Page 15** of the revised manuscript as follows.

“As previously reported, a pre-dose of PEG-HSPC liposomes via intramuscular injection or pre-injection with PEG-based pharmaceutical excipients (poloxamer 188 or poloxamer 407) would induce an obvious lift of anti-PEG antibodies level, eventually boosted blood clearance of the second dose of PEGylated liposomal doxorubicin^{30, 31}. However, where could be the potential destination of the cleared nanoparticles?”

(30) Subasic, C. N.; Butcher, N. J.; Minchin, R. F.; Kaminskas, L. M. Dose-Dependent Production of Anti-PEG IgM after Intramuscular PEGylated-Hydrogenated Soy Phosphatidylcholine Liposomes, but Not Lipid Nanoparticle Formulations of DNA, Correlates with the Plasma Clearance of PEGylated Liposomal Doxorubicin in Rats. *Mol. Pharm.* 2023, 20(7), 3494-3504.

(31) Miao, G.; He, Y.; Lai, K.; Zhao, Y.; He, P.; Tan, G.; Wang, X. Accelerated blood clearance of PEGylated nanoparticles induced by PEG-based pharmaceutical excipients. *J. Control. Release* 2023, 363:12-26.

2. On line 356 of the manuscript, the authors claim that previous studies show limited efficiency of sLip/Dox in comparison to free doxorubicin. However, the stated reference (O'Brien et al, Ann Oncol, 2004) concludes that “[sLip/Dox] provides comparable efficacy to doxorubicin.”

Response: We thank the reviewer for careful reading. The word “limited” was previously used to reflect that PEGylated liposomal doxorubicin was designed to enhance efficacy of doxorubicin but eventually a limited improvement was proved according to the stated reference. **In the revised manuscript, “limited” was substituted by “limited improvement” accordingly as follows (Page 17).**

“Doxil[®] in clinic usually causes an unexpected hand-foot syndrome and displayed **limited improvement** of the overall efficiency compared to the free doxorubicin^{35,36}.”

3. Many of the authors' conclusions are substantiated by repeating experiments in KC-depleted mice as they reveal the critical role of KC. As such, the authors should show the abundance of KC before and after KC depletion.

Response: We thank the reviewer for this suggestion. Kupffer cells (KC) were proved to be crucial for the intrahepatic fate of PEGylated liposomal doxorubicin (sLip/Dox) in our work, and KC-depletion mice model was established using the previously reported method (PMID: 30990673). Meanwhile, the KC abundance in liver was characterized by flow cytometry and the data were shown in **Figure 1D** of the revised manuscript (**Page 6**), which **verified complete KC depletion after treatment.**

Figure 1D. Flow cytometry diagram of nonparenchymal cells in mice liver. The KC was depleted by intravenously injected clodronate liposomes at a volume of 200 μ L and cellular composition was gated at 2 days after injection. The cell types were defined as KC (F4/80⁺), LSEC (CD146⁺).

4. Is there a redistribution of lipid-based nanoparticles from Kupffer cells to LSEC? What is the consequence/effect of the of lipid-based nanoparticle uptake by liver sinusoidal endothelial cells (see 12h time point)? Is the uptake by LSECs zonation dependent? Since Kupffer cells concentrate in the periportal regions, is there more uptake by LSEC closer to the central veins?

Response: According to the current results, it is inclusive if there is a redistribution of liposomal doxorubicin from KC to LSEC, and it is difficult to track the probable transport of liposomes from KC to LSEC as KC would degrade the captured liposomes. Besides, this question is also unsolved for inorganic nanoparticles. Notably, we consider there would be few redistributions of liposomes encapsulated drug from KC to LSEC. As in free Dox treated group (**Figure 2B, Page 9**), there was no detected Dox in LSEC from 1 h to 24 h after injection. While for liposomal Dox, which would be massively captured by KC and degraded to extracellularly release Dox in free form, there would be also nearly no Dox transport into LSEC. The data shown in **Figure 2B** may indicate delayed liposome uptake by LSEC, but not transferred from KC.

Figure 2. Intrahepatic cellular transport of doxorubicin (Dox) in PEGylated liposomes (sLip/Dox), non-PEGylated liposomes (Lip/Dox) or in free form. Dox concentration in Kupffer cells (KC, **A**), liver sinusoidal endothelial cells (LSEC, **B**) at 1 h, 4 h, 12 h and 24 h post intravenous injection of liposomal Dox or free Dox at a dose of 5 mg/kg Dox.

Considering liposomal doxorubicin distribution in LSEC was much less than that in KC (See **Figure 2A-B** of the revised manuscript), we did not investigate the contribution of LSEC in intrahepatic fate of liposomal doxorubicin. As for the probable consequence of liposomes uptake by

LSEC, we considered that most liposomes would be degraded to release the payload (In **Figure 1B-C**, **Page 6-7**, intact liposomes were difficult to transport into hepatocytes).

Figure 1. (B) Mean fluorescence intensity (MFI) variations and **(C)** the positive ratios of major hepatic cells interacting with 5-carboxyfluorescein (FAM) labeled PEGylated liposomes (FAM-sLip). The MFI variations were calculated as increasing folds compared to untreated cells. The positive ratios of untreated cells were gated as less than 0.1%. KC, LSEC, hBC, hTC, pHC and cHC indicated Kupffer cells, liver sinusoidal endothelial cells, intrahepatic B lymphocytes and T lymphocytes, periportal hepatocytes and pericentral hepatocytes, respectively. FAM-sLip was intravenously injected at a dose of 50 mg/kg HSPC.

To explore if the uptake of liposomes by LSEC was zoned, we isolated zoned LSEC using the biomarker CD117 (PMID: 30222169). From the results (**Figure R1**), there was no significant difference in both MFI variations or positive cell ratios, confirming that the uptake of 5-carboxyfluorescein (FAM) labeled PEGylated liposomes (FAM-sLip, in which FAM was chemically conjugated with liposomes) by LSEC was comparable in those two zoned LSEC. As for Kupffer cells distribution in liver, according to **Figure R2**, KC (green) also spread in the pericentral regions after injection of sLip/Dox. As a result, we considered zoned distribution of liposomal doxorubicin in hepatocytes would be more likely from KC instead of LSEC.

Figure R1. FAM labeled PEGylated liposomes (FAM-sLip) distribution in zoned liver sinusoidal endothelial cells (LSEC). **(A)** Gating strategy of fluorescence activated cell sorting for zoned LSEC. The parameters of FSC-A and FSC-H were used for selection of singlets. FSC-A and SSC-A were used for size selection of nonparenchymal cells (NPC). According to the surface biomarker

CD117 levels, LSEC was defined as periportal LSEC (pLSEC) and pericentral LSEC (cLSEC). **(B)** Median fluorescence intensity (MFI) variations and the positive ratios of zonated LSEC treated by FAM-sLip at a dose of 50 mg/kg HSPC. The positive ratios of untreated cells were gated as less than 0.1%. The statistical significance was analyzed by 2-way ANOVA multiple comparisons corrected by Sidak's test. Data are means \pm SDs (n = 3).

Figure R2. Fluorescence images of liver treated by sLip/Dox. The liver cells were labeled as following: **(A)**, DAPI (blue), cell nucleus; glutamine synthetase (green), pericentral hepatocytes; **(B)** DAPI (blue), cell nucleus; E-cadherin (red), periportal hepatocytes; F4/80 (green), Kupffer cells. The mice were intravenously injected with sLip/Dox at a dose of 5 mg/kg Dox. The fluorescence signal from Dox was too weak to be displayed. CV, central vein; PN, portal node. Scale bar, 100 μ m.

5. Figure 4: the authors show metabolite / bile acid data. What is the effect of the gut microbiota?

Response: Previously it was reported that a strain of *Raoultella planticola* in the human gut serving as a potent inactivator under anaerobic conditions to cause doxorubicin degradation via a reductive deglycosylation mechanism (PMID: 29160065), which was discussed in the manuscript as follows (**Page 13** of the revised manuscript). Besides, according to the results in **Figure 4D-E (Page 14-15)**, the metabolite 7-Deo exhibited enhanced distribution in feces in comparison to doxorubicin, which

testified the degradation of doxorubicin due to the gut microbiota.

“As a result, an unexpected large amount of 7-Deo in feces was probably due to metabolism in intestine with the aid of *Raoultella planticola* (*R. planticola*)²⁶”

Figure 4. The feces distribution of Dox (D) and 7-Deo (E) at 12 h and 24 h post injection. The sLip/Dox or free Dox was intravenously injected to C57BL/6 mice at a dose of 5 mg/kg Dox. The statistical significance was analyzed by 2-way ANOVA multiple comparisons corrected by Sidak's test. Data are means \pm SDs (n = 3).

6. In the Significances section, the authors assert liposome encapsulation slows doxorubicin metabolism and excretion. The authors should bolster this point by doing a statistically comparison of doxorubicin concentrations in sLip/Dox treated mice at the 12 h and 24 h timepoints in Figures 4D, Figure 4E, and Figure 4F.

Response: We thank the reviewer for this suggestion. The comparison has been described in the revised manuscript (Page 13) as follows.

“As shown in Figure 4D, Dox distribution was over 3 times more ($p < 0.01$) in feces from free Dox treated mice than that from sLip/Dox treated mice at 12 h and 24 h after administration.”

“As shown in Figure 4E, 7-Deo in feces was much more than Dox in both groups ($p < 0.001$ at 12 h), and exhibited an increasing trend in sLip/Dox treated mice from 12 h to 24 h, while reversed in free Dox treated mice.”

“As shown in Figure 4F, for free Dox treated mice, the Dox concentration in urine was 881.1 ng/mL and 357.2 ng/mL at 12 h and 24 h post injection, about 8.6 ($p < 0.001$) and 1.3 times more than sLip/Dox treated mice, respectively.”

7. The authors found that doxorubicin concentration in KC increased after stimulating mice with sLip. The authors postulate that this finding could be caused by anti-PEG antibodies mediated Fc receptor dependent capture. It is important to show the titers of anti-PEG IgM and IgG at different timepoints could give more context to the findings in Figure 5.

Response: We thank the reviewer for this suggestion. We believe that the similar titers of anti-PEG IgM in mice could be important to evaluate the effects of anti-PEG antibody on intrahepatic cellular distribution of sLip/Dox. We established a standard procedure of anti-PEG antibody pre-existing

mice model in our previous work (PMID: 37845342) by stimulating naïve mice with intravenous injection of PEGylated liposomes (sLip, without drug loading) at a HSPC dose of 5 mg/kg five days before treatment with sLip/Dox. After detection by ELISA to ensure obtained mice at the comparable level of pre-existing anti-PEG IgM (Figure S14 of the revised version), the mice were injected with sLip/Dox at the same time and further evaluation was implemented at different timepoints on different mice. As a result, sLip/Dox suffered similar anti-PEG antibodies at the beginning and the potential variations of anti-PEG antibodies level in treated mice during experiment was not detected considering the same housing and experimental environment.

Figure S14. Anti-PEG IgM level of mice pretreated with sLip at a dose of 5 mg/kg HSPC. The serum from pretreated mice was extracted 5 d after single dose of intravenous injected sLip, and the anti-PEG IgM level was detected by an ELISA method. Data are means \pm SDs (n = 5 for stimulated mice while n = 2 for normal mice).

8. The discussion is overly long and repetitive. A description of the detection of PEG by ELISA should be included in the Experimental Section.

Response: We are grateful to the reviewer for this suggestion. The discussion part has been revised to be more concise. And the detection of anti-PEG antibody by ELISA has been supplemented in the **Methods** section of the revised manuscript (Page 22) as follows.

“Briefly, medium binding 96-well plate was coated with DSPE-PEG₂₀₀₀ (2 μ g/well) in ethanol overnight. After blocking with 5% BSA in 10 mM PBS at 37°C for 1 h and washed with 0.1% Tween-PBS (PBST), 100 μ L mice plasma with serial dilutions by 0.1% BSA-PBS was added into plate and incubated for 1 h at 37°C. After washing with PBST, diluted HRP labeled anti-mice IgM antibody was added with further 1-h incubation at 37°C. Then TMB was added for 8 min chromogenic reaction and the reaction was terminated by 0.18 M sulfuric acid. The OD_{450 nm} was measured by microplate reader.”

9. On line 26 of the Abstract, the authors report that free Dox “quickly distributed into HC without zonated characteristics.” However, Figure 2C does show some differences by liver zonation.

Response: Thanks for your reminding. In fact, “zonated characteristics” for Dox distribution in hepatocytes from different lobule zones in this work referred to Dox accumulation in cHC was significantly higher than that in pHC, which only existed in sLip/Dox treated hepatocytes. However, free Dox treatment did not display obvious trend of the zonated distribution in hepatocytes. Besides, we also stated this phenomenon in the revised article as follows.

In **Abstract**, “It was intriguing that Kupffer cells (KC) initiated and dominated intrahepatic capture of sLip/Dox, and tended to release Dox (from intracellular degraded sLip/Dox) to facilitate **zonated increasing distribution** in hepatocytes (HC) along the lobule porto-central axis”.

In **Page 8**, “Besides, in liposomal Dox treated mice, Dox distribution in HC from different lobule zones also displayed a zonated distribution profile from cHC to pHC, **as Dox concentration in pericentral HC was roughly much higher than that in HC from the other two regions**”.

10. Line 451: How can a lower plasma level of anti-PEG antibodies be achieved in patients?

Response: To lower plasma level of anti-PEG antibodies in patients could be complicated. However, screening patients with lower pre-existing anti-PEG antibodies level in clinic, or coating PEGylated liposomal drug with PEG-scFv to circumvent anti-PEG antibodies (PMID: 33383098), could efficiently extend blood circulation of PEGylated liposomal drug. We also revised the illustration as follows in **Page 19-20**.

“Besides, for precision medication of PEGylated liposomal doxorubicin in clinic, as lifted anti-PEG antibodies level was widely detected⁴⁵ and would eventually upregulate doxorubicin accumulation in hepatocytes and following excretion into the intestine, strictly controlled dosage should be necessary for patients with higher pre-existing anti-PEG antibodies to avoid probable toxicity of doxorubicin to liver or intestine.”

Reviewer #2 (Remarks to the Author):

Jiang et al has reported the role of Kupffer cell (KC) in intrahepatic accumulation of drug through PEGylated liposomal doxorubicin (sLip/Dox). The report has shown KC dominated intrahepatic capture of sLip/Dox and released the intracellular payload Dox to facilitate zoned distribution in hepatocytes (HC). The report also showed a size dependent KC capture ability of sLip/Dox by increasing the size of liposomes size. The report also found the similar effect in the presence of anti-PEG antibodies. However, intravenously injected free Dox failed to do. The study is systematic, but the results are very obvious as shown by others before.

Major:

1. The novelty of this report is completely missing. This falls in the ‘yet another’ category with liposomal interaction with KC. Kupffer cells are very well known for uptaking liposomes. In 1980’s, Roerdink et al showed in their seminal paper ‘Interaction of liposomes with hepatocytes and Kupffer cells’. More recently, as reported by Rooijen et al ‘Kupffer cell depletion by liposome-delivered drugs: Comparative activity of intracellular clodronate, propamidine, and ethylenediaminetetraacetic acid’ (<https://doi.org/10.1002/hep.510230544>). In fact, it is becoming a strategy that Tavares et al, showed the ‘Effect of removing Kupffer cells on nanoparticle tumor delivery’ due to the same reason (<https://doi.org/10.1073/pnas.1713390114>). The authors, therefore, need to clearly redefine the novelty aspect of the present report.

Response: We thank the reviewer for the suggestions. In fact, there were quite a lot of publications about nanoparticle behaviors in liver up to now. As mentioned, the first article (PMID: 6724117) provided information about initial liposomes or payload distribution in several liver cells based on radioactively labelled compositions. The second article (PMID: 8621159) proved clodronate liposomes were efficient for KC depletion in liver. The third article (PMID: 29208719) focused on depleting KC to increase nanoparticle distribution in solid tumor, **but overlooking the further process in liver**. Recently there were also valuable reports about intrahepatic process of nanoparticles (PMID: 30990673; PMID: 27525571), **but still be confined to inorganic nanoparticles and without consideration in the payload**.

To establish a detailed blueprint about intrahepatic fate of lipid-based nanomedicine, in this work, not only the key role of KC in intrahepatic transport of PEGylated liposomal doxorubicin was defined, but **more importantly, the following process after KC capture was illustrated**. Based on the results, the major form of drug into HC, zoned drug distribution in HC, and the effects of physicochemical properties of liposomes (sizes, surface modifications, etc.) or physiological

environment (Kupffer cells, anti-PEG IgM, etc.) on intrahepatic transport/distribution/metabolism/excretion were also uncovered, and we believe that all the information would be crucial for future development or clinical usage of liposomal drug but still missing before our work.

2. It is obvious therefore that sLip/Dox will interact with Kupffer cells and will increase the hepatic cell delivery as shown in Figure 1B-E and with depletion in Figure 1C-D. Is there anything else that the authors were expected here? If so, it is partly clear.

Response: In this work, we endeavored to establish the whole intrahepatic fate and clarify the regulation mechanism on PEGylated liposomal doxorubicin behaviors. As shown in **Figure 1B-E** (Page 6-7) and **Figure 3C-D** (Page 11), KC exhibited powerful capture capacity on intact liposomes as mentioned. Besides, it was also revealed the distinct uptake behaviors for intact liposomes by the other cells, containing lagging capture by LSEC, more uptake by intrahepatic T lymphocytes than B lymphocytes, and trace of intact liposomes transport into HC, without reported before. Also taking results in **Figure 2** (Page 9) into consideration, **different intrahepatic traffic between liposomes and the encapsulated payload was displayed**, as intact liposomes would be impeded by hepatic nonparenchymal cells to reach the parenchyma, but would be degraded to release the payload into HC, which was still obscure before.

Figure 1. (B) Mean fluorescence intensity (MFI) variations and **(C)** the positive ratios of major hepatic cells interacting with 5-carboxyfluorescein (FAM) labeled PEGylated liposomes (FAM-sLip). The MFI variations were calculated as increasing folds compared to untreated cells. The positive ratios of untreated cells were gated as less than 0.1%. KC, LSEC, hBC, hTC, pHC and cHC indicated Kupffer cells, liver sinusoidal endothelial cells, intrahepatic B lymphocytes and T lymphocytes, periportal hepatocytes and pericentral hepatocytes, respectively. FAM-sLip was intravenously injected at a dose of 50 mg/kg HSPC. **(D)** Flow cytometry diagram of nonparenchymal cells in mice liver. The KC was depleted by intravenously injected clodronate

liposomes at a volume of 200 μ L and cellular composition was gated at 2 days after injection. The cell types were defined as KC (F4/80⁺), LSEC (CD146⁺). (E) Tissues distribution of doxorubicin encapsulated in liposomes (90 nm) at 4 h or 12 h post injection with KC depletion or not. The PEGylated liposomal Dox (sLip/Dox) was intravenously injected at a dose of 5 mg/kg Dox. The statistical significance was analyzed by 2-way ANOVA multiple comparisons corrected by Sidak's test. Data are means \pm SDs (n = 3).

Figure 2. Intrahepatic cellular transport of doxorubicin (Dox) in PEGylated liposomes (sLip/Dox), non-PEGylated liposomes (Lip/Dox) or in free form. Dox concentration in Kupffer cells (KC, **A**), liver sinusoidal endothelial cells (LSEC, **B**) and zoned hepatocytes (HC, **C-F**) at 1 h, 4 h, 12 h and 24 h post intravenous injection of liposomal Dox or free Dox at a dose of 5 mg/kg Dox. Rough calculation of total Dox distribution in major liver cells from mice treated by sLip/Dox (**G**), Lip/Dox (**H**) or free Dox (**I**). The Dox concentration was reused from data in **A-F**. The %liver cells ratio referred to Ref. 18, as 6% for KC, 15% for LSEC, and 23.3% for cHC, mHC, pHC, respectively, considering a total HC occupied about 70% of the total liver cells. cHC, pericentral hepatocytes; mHC, mid-lobule hepatocytes; pHC, periportal hepatocytes. The statistical significance was analyzed by 2-way ANOVA multiple comparisons corrected by Sidak's or Tukey's test. Data are means \pm SDs (n = 3).

Figure 3. The sLip with different particle sizes (90 nm, 120 nm or 300 nm) encapsulated Dox distribution in LSEC (C-D) at 4 h or 12 h post injection with KC depletion or not. The sLip/Dox was intravenously injected at a dosage of 5 mg/kg doxorubicin. The statistical significance was analyzed by 2-way ANOVA multiple comparisons corrected by Sidak's test. Data are means \pm SDs (n = 3).

3. The size dependency also is expected due to the phagocytosis cut-off size. It would have been interesting to see if there are any lower and upper cut-off limits exists for the KC uptake.

Response: The size dependency of KC uptake on liposomes should be interesting, and we also made more efforts to investigate the effects on following intrahepatic fate of the encapsulated drug. As a result, we compared intrahepatic cellular distribution of liposomes with particle sizes of 90 nm, 120 nm and 300 nm (as shown in Figure 3B, Page 11 of the revised manuscript). Comparing doxorubicin distribution in KC, the largest one (300 nm) induced more doxorubicin distribution in KC, similar to previous results in gold nanoparticles (PMID: 30990673), revealing that KC preferred to sequester large nanoparticles than the small ones (ranged from 4-200 nm for gold nanoparticles as previously reported, but 90-300 nm in our work). Besides, with more capture by KC for the large liposomal doxorubicin, there was still much less Dox distribution in LSEC for the three sizes of liposomal doxorubicin (Figure 3C-D), but Dox accumulation in HC significantly increased (Figure 3G-H), also with a zoned distribution along the porto-central axis.

We also supplement content about the effects of particle sizes on intrahepatic process of sLip/Dox as follows.

“As for intracellular distribution, sLip/Dox with larger size accumulated in KC significantly more than the smaller one, as the larger sLip/Dox induced 1.5 to 3 folds higher Dox accumulation than the small counterpart (Figure 3B). In LSEC, there was a sudden decrease for sLip/Dox accumulation compared to that in KC, ranging from 1 to 4 ng/10⁶ cells in KC-normal state (Figure 3C-D). Notably, for 90 nm sLip/Dox treated LSEC with KC depletion at 4 h, Dox concentration increased to 3.6 ng/10⁶ cells, significantly more than in the normal condition, revealing a probable shielding effect of KC on LSEC for liposomes capture in short time. However, the situation reversed for 300 nm sLip/Dox, as Dox concentration in LSEC with KC depletion was 2.2 times lower than

that in KC-normal state, probably due to LSEC uptake of nanoparticles with a cut-off size no more than 300 nm.”

“Similar to the results in KC, liposomal Dox with a larger particle size also accumulated more in HC compared to the smaller one (Figure 3G, S10C), especially in cHC, revealing a potential relationship of KC capture and Dox accumulation in HC.”

“Comparing different sizes of liposomal Dox total distribution in liver cells (Figure 3H, S12), Dox accumulation was ranking as KC > LSEC > cHC > mHC > pHC, and increased with enlarged particle sizes, further to testify KC initiated and promoted liposomal Dox zoned distribution in HC.”

Figure 3. The sLip with different particle sizes (90 nm, 120 nm or 300 nm) encapsuled Dox distribution in KC (B) and LSEC (C-D) with KC depletion or not at 4 h or 12 h post injection. The sLip/Dox was intravenously injected at a dosage of 5 mg/kg doxorubicin. The statistical significance was analyzed by one-way ANOVA multiple comparisons corrected by Tukey’s test for B, and by 2-way ANOVA multiple comparisons corrected by Sidak’s test for C-D. Data are means ± SDs (n = 3).

Figure 3. (G) Distribution of sLip/Dox (90 nm, 120 nm or 300 nm) in zonated HC at 4 h post injection in KC-normal condition. The sLip/Dox was intravenously injected at a dosage of 5 mg/kg doxorubicin. The statistical significance was analyzed by one-way ANOVA multiple comparisons corrected by Tukey’s test. Data are means ± SDs (n = 3). (H) Rough calculation of total Dox distribution in major liver cells at 4 h after injection. The %liver cells ratio referred to Ref. 18, as 6% for KC, 15% for LSEC, and 23.3% for cHC, mHC, pHC, respectively, considering a total HC occupied about 70% of the total liver cells.

4. The other concern is about the reason behind using the PEG molecule. This might well be to increase the circulation time. It would have been interesting to see on how the bare non-stealth version behave. Similarly, it is not clear about the role of anti-PEG antibody. If the authors were expected otherwise, then the hypothesis for this experiment is unclear at this stage.

Response: Surface PEGylation of nanoparticles can improve their biocompatibility, extend the blood circulation, and has promoted most nanomedicines in clinical use (PMID: 38193121). PEGylated liposomal doxorubicin (sLip/Dox) is the first FDA-approved anticancer nanomedicine, and still plays an important role in clinic. As a result, we chose sLip/Dox as the model liposomes to clarify the whole intrahepatic process, which could be also valuable for other PEGylated nanomedicines.

As for the role of anti-PEG antibodies, with the long-term use of PEG molecules in the daily life (food, cosmetics, drug, etc.), anti-PEG antibodies have been widely detected in human beings (PMID: 27726379) and would cause unignored effects on PEGylated drug safety and efficacy (PMID: 32745496). However, researches on anti-PEG antibodies mainly focused on accelerated blood clearance effects and mononuclear macrophage system capture, but ignored following process for example accumulation/metabolism in hepatocytes (HC), which largely determined eventual biological effects of PEGylated nanomedicines. As a result, sLip/Dox distribution in liver cells was evaluated under pre-existing antibodies condition (which is similar to the clinical conditions), and the results could be important for the further development and application of PEGylated nanomedicines.

Besides, we also compared intrahepatic behaviors of PEGylated and non-PEGylated liposomal doxorubicin (Lip/Dox) as shown in **Figure 2 (Page 9)**. Concisely, intrahepatic cellular distribution was similar for sLip/Dox and Lip/Dox, except a lagging capture by KC for Lip/Dox. The related content has also been supplemented in the revised manuscript as follows.

“For non-PEGylated Lip/Dox, Dox distribution in KC was hysteretic to the PEGylated one, and displayed an increasing trend from 1 h to 12 h, indicating differences in KC capture of two different populations of liposomes, largely due to existence of PEG-lipid. Both liposomal Dox displayed a slow and moderate distribution behavior into LSEC. After 1 h, no detectable Dox was measured in LSEC, but with continuous increase from 4 h to 24 h, ranging from 0.1 to 2.9 ng/10⁶ cells, which was less than that in KC but significantly higher than free Dox treated LSEC especially after 12 h.”

“More importantly, as Dox distribution in liver cells ranking as KC >> zonated HC for liposomal Dox treated group (**Figure 2G-H**, but reversed in free Dox treated group as shown in **Figure 2I**), indicating an intercellular transport from KC to HC for Dox encapsuled in liposomes.

As sLip/Dox induced a more fast Dox distribution in KC than Lip/Dox, peaking at 1 h or 12 h, respectively, and there was also slower Dox transport to HC in Lip/Dox treated group, also indicating a potential intercellular transport of liposomal Dox from KC to HC.”

Figure 2. Intrahepatic cellular transport of doxorubicin (Dox) in PEGylated liposomes (sLip/Dox), non-PEGylated liposomes (Lip/Dox) or in free form. Dox concentration in Kupffer cells (KC, **A**), liver sinusoidal endothelial cells (LSEC, **B**) and zoned hepatocytes (HC, **C-F**) at 1 h, 4 h, 12 h and 24 h post intravenous injection of sLip/Dox or free Dox at a dose of 5 mg/kg Dox. Rough calculation of total Dox distribution in major liver cells from mice treated by sLip/Dox (**G**), Lip/Dox (**H**) or free Dox (**I**). The Dox concentration was reused from data in **A-F**. The %liver cells ratio referred to Ref. 18, as 6% for KC, 15% for LSEC, and 23.3% for cHC, mHC, pHC, respectively, considering a total HC occupied about 70% of the total liver cells. cHC, pericentral hepatocytes; mHC, mid-lobule hepatocytes; pHC, periportal hepatocytes. The statistical significance was analyzed by 2-way ANOVA multiple comparisons corrected by Sidak's or Tukey's test. Data are means \pm SDs (n = 3).

5. It would have been interesting to see whether the surface charge has any role to play for Kupffer cell uptake. Surface zeta potential of the sLip/Dox might influence the uptake.

Response: We thank the reviewer for this suggestion. We believe that surface zeta potentials of liposomal doxorubicin would influence Kupffer cells (KC) uptake. However, the related research was not implemented as we mainly focused on the subsequent process after KC capture. In this work, it was indicated that more capture of PEGylated liposomes would induce more payload distribution in hepatocytes (HC), and KC capture could also regulate other intrahepatic behaviors for example the zoned drug distribution in HC and drug metabolism. Besides, there would be

various approved nanomedicines with diverse interactions with KC, and our findings, especially that “KC capture could regulate following intrahepatic process of nanomedicine”, may provide potential guidance for other researchers in the related fields.

6. The authors must show some direct evidence on the extent of drug loading capacity of the proposed system.

Response: PEGylated liposomal doxorubicin was well-prepared in laboratory with a standard procedure. Physicochemical properties of prepared liposomal doxorubicin were characterized as in **Table S1**, and drug loading capacity was evaluated as in **Methods** section (**Page 24**). Besides, cryo-TEM pictures of different liposomal doxorubicin were also displayed as in **Figure S5**, which could be clearly seen the loading doxorubicin.

Table S1. Particle sizes, polydispersity index (PDI) and surface zeta potential (ZP) of FAM-sLip and sLip/Dox.

	Size (d. nm)	PDI	ZP (mV)	LC/%	EE/%
FAM-sLip	83.7 ± 0.9	0.037 ± 0.031	-19.4 ± 0.8	-	-
sLip/Dox (90 nm) ^a	87.5 ± 1.0	0.108 ± 0.036	-22.5 ± 1.0	14.5 ± 0.5	98.6 ± 0.5
sLip/Dox (120 nm)	120.7 ± 1.0	0.113 ± 0.012	-16.9 ± 1.6	12.8 ± 0.6	96.0 ± 0.3
sLip/Dox (300 nm)	293.6 ± 11.2	0.132 ± 0.035	-18.7 ± 1.1	7.30 ± 0.2	95.5 ± 0.6
Lip/Dox	102.4 ± 2.1	0.049 ± 0.034	-8.16 ± 0.6	12.4 ± 0.7	96.7 ± 0.5

a. sLip/Dox used in this work was with a particle size of 90 nm except particularly statement.

PDI, polydispersity index; ZP, surface zeta potential; LC, drug loading capacity, calculated by the ratio of doxorubicin concentration to HSPC concentration; EE, encapsulation efficiency, calculated as the ratio of free doxorubicin concentration to total doxorubicin in sLip/Dox.

Figure S5. Cryo-TEM pictures of non-PEGylated liposomal doxorubicin (Lip/Dox) and PEGylated liposomal doxorubicin (sLip/Dox) with different particle sizes. The lipid membrane was labeled by the circle while doxorubicin nanocrystal was labeled by the rectangle. Scale bar, 100 nm.

7. More proper justification of the exact application of this study is needed, as it is not clearly mentioned.

Response: Thanks for your suggestions and the related discussion has been supplemented in **Discussion** section (**Page 19-20**) as follows.

“The results in the present study revealed that regulation on KC capture could be efficient to control *in vivo* behaviors of liposomes. For example, to control liposomes in smaller sizes, or inhibit complement activation, which could lower KC capture of liposomes⁴³, would benefit to avoid excessive intrahepatic drug accumulation, meanwhile reduce potential intrahepatic metabolism burden and even toxicity to cells along the hepatobiliary elimination pathway. Besides, for precision medication of PEGylated liposomal doxorubicin in clinic, as lifted anti-PEG antibodies level has been widely detected⁴⁴ and would eventually upregulate doxorubicin accumulation in hepatocytes and following excretion into the intestine, strictly controlled dose should be necessary for patients with high pre-existing anti-PEG antibodies to avoid probable toxicity of doxorubicin to liver and intestine.”

Reviewer #3 (Remarks to the Author):

This work investigated the fate of liposomes and drugs in the liver and proposed a novel mechanism for the liver processing of liposomes. This is indeed an important topic, which can help the future design of safer and more effective liposomal delivery of drugs. However, upon initial review, some technical and experimental design flaws need to be addressed. More work is needed to support the proposed model of liposome capture and drug metabolism by KC and LSEC. My specific comments are as follows:

1) It is intriguing that with KC depletion, the liver concentration of Dox reduced drastically but concentration in other organs did not change much. Have the authors examined the blood half-life of Dox? Where did the particles go?

Response: Thanks to the reviewer for the kind suggestion. We have compared the blood circulation of sLip/Dox with KC depletion or not. In fact, considering long blood circulation ability of sLip/Dox and the inlet/outlet balance in liver or other organs, Dox concentration in liver or other organs was much lower than blood concentration within 24 h after administration. According to the results in **Figure S6** of the revised supplementary information, compared to that in KC-normal mice, sLip/Dox exhibited an obvious extended blood circulation in KC-depletion mice. As a result, in the KC-depletion state, much more liposomes circulated in blood, waiting for an extended elimination from liver or other sites.

Figure S6. Pharmacokinetic profiles of sLip/Dox, Lip/Dox and free Dox in C57BL/6 mice. The mice were intravenously injected with different preparations at a dose of 5 mg/kg Dox. Data are means \pm SDs (n = 3).

2) The authors used ng/10⁶ cells as the concentration of drug to speculate the mechanism of drug distribution. However, the absolute number of cells in each liver should also be considered. It is possible that one cell population is very abundant, so even when the concentration calculated as ng/10⁶ cells is low, it's overall uptake might be more dominant.

Response: We thank the reviewer for this insightful suggestion. In this work, we mainly compared Dox concentration in KC, LSEC and zonated HC, and those cells number was no more than 10 folds to each other as previously reported (PMID: 16447271). To make it more intuitive to catch Dox distribution in different cells, the relevance of absolute cells number and Dox distribution in those cells were displayed as in **Figure 2G-I (Page 9)**, **3H (Page 11)** of the revise manuscript and **S12** of the revised supplementary information, which all displayed intracellular drug distribution ranking as $KC > LSEC > \text{zonated HC}$ for liposomal doxorubicin treated mice.

Figure 2. Rough calculation of total Dox distribution in major liver cells from mice treated by sLip/Dox (G), Lip/Dox (H) or free Dox (I). The Dox concentration was reused from data in A-F. The %liver cells ratio referred to Ref. 18, as 6% for KC, 15% for LSEC, and 23.3% for cHC, mHC, pHC, respectively, considering a total HC occupied about 70% of the total liver cells.

Figure 3H. Rough calculation of total Dox distribution in major liver cells at 4 h after injection. The %liver cells ratio referred to Ref. 18, as 6% for KC, 15% for LSEC, and 23.3% for cHC, mHC, pHC, respectively, considering a total HC occupied about 70% of the total liver cells.

Figure S12. Rough calculation of total Dox distribution in major liver cells at 12 h after injection. Data were calculated as in Figure 3H.

3) One major problem of methodology is that fluorescent signal was used to quantify cellular uptake of FAM-sLip in figure 1 and mass spec concentration of Dox was used to quantify uptake of sLip/Dox. The difference of kinetics of the two sLip could be due to the sensitivity of two different testing methods. I recommend the authors to be consistent with quantification methodology, for example, also conjugate FAM to Doxil.

Response: In this work, FAM was chemically conjugated to sLip to quantify PEGylated liposomes distribution, while Dox concentration detected by mass spectrometry was utilized to evaluate intrahepatic distribution of the payload mediated by PEGylated liposomes. In other words, those two quantification methodologies were used for evaluation of liposomes and the payload distribution, respectively, and reliable to support the research purpose. Besides, considering complicated biofluids composition and different fluorescence properties of free Dox and encapsulated Dox, quantification by fluorescence could be less precise compared to the mass spec methodology.

4) More data is needed to support the hypothesis that Dox was released from liposomes in KC and infuse into HC. The decrease concentration of Dox in KC can be achieved without liposomal release. It is also possible that Dox has been degraded or Lip/Dox was released in whole.

Response: We considered sLip/Dox captured by KC more likely to release Dox in free form, which subsequently entered HC through the fenestration in liver sinusoid endothelium mainly based on the following two reasons:

(1) Intracellular concentration of Dox in KC was much higher than that in HC; however, there was no metabolite detected in KC but displayed zonated distribution in HC as shown in **Figure 4A** (Page 14). As a result, it would be difficult for Dox degradation in KC.

Figure 4A. Metabolic kinetics of Dox to Doxol in zoned hepatocytes from sLip/Dox treated mice. The sLip/Dox was intravenously injected to C57BL/6 mice at a dosage of 5 mg/kg Dox. The statistical significance was analyzed by 2way ANOVA multiple comparisons corrected by Sidak's test. Data are means \pm SDs (n = 3).

(2) According to the results in **Figure 1B-C** (Page 6-7), FAM-sLip distribution in HC was much less than distribution in nonparenchymal cells. Besides, when KC was depleted, Dox distribution in liver or HC obviously decreased as shown in **Figure 1E** (Page 6-7), **3E** (Page 11),

also with extended blood circulation as in **Figure S6**. From the above results, it was deduced that intact sLip/Dox were difficult to enter HC, and the pretreatment of sLip/Dox in KC was indispensable for Dox accumulation in HC, namely sLip/Dox degraded in KC to release Dox in free form and entered liver parenchyma.

Figure 1. Intrahepatic distribution of PEGylated liposomes. (A) Intravenously injected PEGylated liposomes interact with various hepatic cells during flowing through the liver sinusoid. (B) Mean fluorescence intensity (MFI) variations and (C) the positive ratios of major hepatic cells interacting with 5-carboxyfluorescein (FAM) labeled PEGylated liposomes (FAM-sLip). The MFI variations were calculated as increasing folds compared to untreated cells. The positive ratios of untreated cells were gated as less than 0.1%. KC, LSEC, hBC, hTC, pHC and cHC indicated Kupffer cells, liver sinusoidal endothelial cells, intrahepatic B lymphocytes and T lymphocytes, periportal hepatocytes and pericentral hepatocytes, respectively. FAM-sLip was intravenously injected at a dose of 50 mg/kg HSPC. (D) Flow cytometry diagram of nonparenchymal cells in mice liver. The KC was depleted by intravenously injecting clodronate liposomes at a volume of 200 μ L and cellular composition was gated at 2 days after injection. The cell types were defined as KC (F4/80+), LSEC (CD146+). (E) Tissues distribution of doxorubicin encapsulated in liposomes (90 nm) at 4 h or 12 h post injection with KC depletion or not. The PEGylated liposomal Dox (sLip/Dox) was intravenously injected at a dose of 5 mg/kg Dox. The statistical significance was analyzed by 2-way ANOVA multiple comparisons corrected by Sidak's test. Data are means \pm SDs (n = 3).

Figure 3E. Distribution of sLip/Dox (90 nm) in zoned HC at 4 h post injection with KC depletion or not.

Figure S6. Pharmacokinetic profiles of sLip/Dox, Lip/Dox and free Dox in C57BL/6 mice. The mice were intravenously injected at a dose of 5 mg/kg Dox. Data are means \pm SDs (n = 3).

5) The majority of the results are bar graphs of drug concentration. I recommend the authors to add some immune fluorescence images of different liver cell populations and drug distribution at different time point.

Response: Thanks for your suggestions. In fact, we have tried to display Dox distribution in liver cells (different cells could be separated well by surface markers as shown in **Figure R2**) by other form for example fluorescence images. However, Dox encapsulation in liposomes or in free form with a definitely different fluorescence signal, as free Dox emitted much stronger signal than the encapsulated form when detecting by the fluorescent method. Besides, as for liver cryo-sections, fluorescence signal from Dox was too weak to separate from the background.

Figure R2. Fluorescence images of liver treated by sLip/Dox. The liver cells were labeled as following: **(A)**, DAPI (blue), cell nucleus; glutamine synthetase (green), pericentral hepatocytes; **(B)** DAPI (blue), cell nucleus; E-cadherin (red), periportal hepatocytes; F4/80 (green), Kupffer cells. The mice were intravenously injected with sLip/Dox at a dose of 5 mg/kg Dox. The fluorescence signal from Dox was too weak to be displayed. CV, central vein; PN, portal node. Scale bar, 100 μm .

6) Doxil by itself is toxic and can affect function of liver cells. Please add a control experiment to exclude the effects of cytotoxicity of Doxil.

Response: We believed that the toxicity of doxorubicin could affect function of liver cells. In this work, we injected free Dox or sLip/Dox at a single dosage of 5 mg/kg doxorubicin, which also has been applied in various publications. Besides, blood routine examination and liver function evaluation by serum biochemical analysis at 24 h after sLip/Dox injection were supplemented in the revised version as follows (**Page 7**).

“Besides, as shown in **Figure S7**, blood cell counts were comparable between untreated and sLip/Dox treated mice, except for lower number of white blood cells in treated mice, but still in the normal range ($3.2\text{-}12.7 \times 10^3/\mu\text{L}$). To evaluate liver function after treatment by sLip/Dox, serum biochemical analysis was implemented and indicated no significant difference compared to that in untreated mice (**Figure S8**). From the hematoxylin-eosin staining sections of liver in untreated or treated mice (**Figure S9**), there was also no obvious toxicity as both nucleus and cytoplasm were

clearly stained. From the above results, single injection of sLip/Dox at a dosage of 5 mg/kg Dox would induce no toxicity for liver function or cell counts, supporting to following evaluation by this dosage.”

Figure S7. Routine blood tests of the peripheral blood from untreated mice and mice treated by sLip/Dox. The treated mice were intravenously injected with sLip/Dox at a dosage of 5 mg/kg doxorubicin, and the whole blood was gathered at 24 h after injection in EDTA-containing tube at room temperature before test. The statistical significance was analyzed by two-tailed unpaired t-test. Data are means ± SDs (n = 3). WBC, white blood cell; RBC, red blood cell; PLT, platelet; HGB, hemoglobin; HCT, hematocrit; NEUT, neutrophil; LYMPH, lymphocyte; MONO, monocyte; EOS, eosinophilic; BASO, basophil; MCV, mean corpuscular volume; RDW, red cell distribution width; CH, corpuscular hemoglobin; MCH, mean corpuscular hemoglobin; MCHC, mean corpuscular

hemoglobin concentration; HDW, hemoglobin distribution width; MPV, mean platelet volume; PDW, platelet volume distribution width; PCT, platelet crit.

Figure S8. Liver function evaluation reflecting by serum biochemical analysis. The treated mice were intravenously injected with sLip/Dox at a dosage of 5 mg/kg doxorubicin, and the whole blood was gathered at 24 h after injection in tube at room temperature before test, further to keep at room temperature for 1 h to obtain the serum. The statistical significance was analyzed by two-tailed unpaired t-test. Data are means \pm SDs (n = 3). ALB, albumin; ALP_2c, alkaline phosphatase 2c; ALT, alanine aminotransferase; AST, aspartate aminotransferase; CHE, cholinesterase; TP, total protein; GGT, glutamyltransferase; TBIL_2, total bilirubin 2.

Figure S9. Hematoxylin-eosin staining sections of liver or intestine. Mice in sLip/Dox treated group were intravenously injected with sLip/Dox at a dosage of 5 mg/kg doxorubicin. The tissues were gathered 24 h after injection. Scale bar, 50 μ m.

7) KC and LSEC share several functions and gene expression signatures. Recent research also identified there might be subsets of KC exist that assemble the features of LSEC (doi.org/10.1016/j.cell.2021.12.018). This work distinguishes KC and LSEC with just F4/80

and CD146, which might be over-simplified. More specific characterization of these two populations is highly recommended.

Response: This is a valuable suggestion to improve this work. In fact, with the development in spatial proteogenomics, much higher resolution about cell heterogeneity analysis has been achieved recently, which would promote a more precise evaluation about contribution among the different cell subsets for intrahepatic fates of nanoparticles. However, up to now, some technologies such as single-cell CITEseq are still difficult to be utilized for intracellular drug concentration evaluation considering cells are lysed to release both biomarkers and drug before cell subsets identification. Besides, F4/80 and CD146 are constantly typical surface markers for KC and LSEC separation, respectively, as reported in several articles (*Cell. Metab.* 2014, 20(6): 1076-87; *Nature* 2021, 589(7840):131-6).

REVIEWERS' COMMENTS

Reviewer #2 (Remarks to the Author):

The authors have addressed most of the concerns and justified the points rationally.

Reviewer #3 (Remarks to the Author):

The authors have done a good job addressing my concerns. I have no additional request.